# Integrating single cell expression quantitative trait loci summary statistics to understand complex trait risk genes

Lida Wang [1,8], Chachrit Khunsriraksakul [2,3,8], Havell Markus[2,3], Dieyi Chen[1], Fan Zhang [2], Fang Chen[1], Xiaowei Zhan[4,5,6], Laura Carrel [7] ✉, Dajiang. J. Liu [1,2,4] ✉ & Bibo Jiang [1] ✉

Transcriptome-wide association study (TWAS) is a popular approach to dissect the functional consequence of disease associated non-coding variants. Most existing TWAS use bulk tissues and may not have the resolution to reveal cell-type specific target genes. Single-cell expression quantitative trait loci (sc-eQTL) datasets are emerging. The largest bulk- and sc-eQTL datasets are most conveniently available as summary statistics, but have not been broadly utilized in TWAS. Here, we present a new method EXPRESSO (EXpression PREdiction with Summary Statistics Only), to analyze sc-eQTL summary statistics, which also integrates 3D genomic data and epigenomic annotation to prioritize causal variants. EXPRESSO substantially improves existing methods. We apply EXPRESSO to analyze multi-ancestry GWAS datasets for 14 autoimmune diseases. EXPRESSO uniquely identifies 958 novel gene x trait associations, which is 26% more than the second-best method. Among them, 492 are unique to cell type level analysis and missed by TWAS using whole blood. We also develop a cell type aware drug repurposing pipeline, which leverages EXPRESSO results to identify drug compounds that can reverse disease gene expressions in relevant cell types. Our results point to multiple drugs with therapeutic potentials, including metformin for type 1 diabetes, and vitamin K for ulcerative colitis.

Genome-wide association studies have revealed numerous variants associated with many human diseases and traits[1]. Most identified associations are non-coding and influence disease risk via their regulatory effects on gene expression, which can be tissue and cell-type specific[2,3]. Interpreting the functional consequence of non-coding variants is challenging and requires integrating functional genomic data from disease-relevant cell types and tissues.

Recently, transcriptome-wide association study (TWAS) has become a popular gene-based association analysis method for understanding non-coding variants. Many studies have applied TWAS to identify risk genes for complex human diseases[4–7]. Briefly, TWAS

[1]Department of Public Health Sciences; Pennsylvania State University College of Medicine, Hershey, Pennsylvania, USA. [2]Bioinformatics and Genomics PhD Program; Pennsylvania State University College of Medicine, Hershey, Pennsylvania, USA. [3]Institute for Personalized Medicine; Pennsylvania State University College of Medicine, Hershey, Pennsylvania, USA. [4]Department of Statistical Science, Southern Methodist University, Dallas, TX, US. [5]Quantitative Biomedical Research Center, Department of Population and Data Sciences, University of Texas Southwestern Medical Center, Dallas, TX, US. [6]Center for Genetics of Host Defense, University of Texas Southwestern Medical Center, Dallas, TX, US. [7]Department of Biochemistry and Molecular Biology; Pennsylvania State University College of Medicine, Hershey, Pennsylvania, USA. [8]These authors contributed equally: Lida Wang, Chachrit Khunsriraksakul. ✉e-mail: lcarrel@pennstatehealth.psu.edu; dajiang.liu@psu.edu; bjiang@phs.psu.edu

first derives gene expression prediction models from datasets with matched genotypes and gene expression data. These models are utilized to impute gene expression levels into GWAS datasets, which are tested for associations with complex traits to identify significant gene-trait associations (GTAs).

Gene regulatory mechanisms can be cell-type specific, and causal variants may function and influence disease outcomes only in certain cell types. To date, most TWAS methods use RNASeq data from bulk tissues, including multiple cell types in different proportions. As a result, measured gene expression levels reflect weighted averages across different cell types. Therefore, eQTL and TWAS analysis based on bulk tissues only provide limited granularity and may fail to reveal causal effects present only in a subset of cell types, particularly if that cell type is rare. Thanks to the decreasing costs of data generation, larger sc-RNASeq datasets are emerging. Consortia efforts are also underway to aggregate multiple datasets across studies and release sc-eQTL summary statistics. Most current studies seek to integrate sc-eQTL data with GWAS using colocalization[8–10]. No studies, to our knowledge, have sought to integrate sc-eQTL via TWAS. In contrast to colocalization, TWAS seeks to understand causal pathways from genetic variants → predicted gene expression → phenotype. This allows us to estimate the effects of predicted gene expression on phenotypes and characterize how gene expression mediates the effects of regulatory variants. In addition, TWAS, as an association method, can identify novel gene-level associations and reveal risk genes, which is something that colocalization does not do. Extending TWAS methods to exploit large sc-eQTL datasets will further allow us to characterize the effect size heterogeneity of predicted gene expression in different cell types[11,12], an important but unanswered question, and identify novel loci to gain biological and clinical insights.

In this work, we present a new TWAS method EXPRESSO (EXpression PREdiction with Summary Statistics Only). Unlike most existing TWAS methods that require individual-level genotype and expression information[3–6,13,14], EXPRESSO can effectively use the largest datasets of sc- or bulk-RNASeq summary statistics, e.g., the ones from eQTLGen[15] or sc-eQTLGen[16]. It also utilizes epigenomic and 3D genomic information to prioritize putative functional cis-regulatory variants and improve power. We compared EXPRESSO against existing TWAS methods that rely on individual-level genotype and phenotype data and against polygenic risk score methods adapted to analyze eQTL datasets. We demonstrate substantial improvement in TWAS's power in simulation and applied data analysis. We apply the methods to eQTLGen and sc-eQTLGen summary statistics and 14 autoimmune diseases with a maximum sample size of 728,548 to discover novel cell-type specific gene-level associations and identify drugs that we may repurpose to treat these disorders. We also develop a novel heterogeneity statistic to rigorously assess how the effects of genetically predicted gene expressions vary between cell types.

## Results
### EXPRESSO method overview
In this section, we first describe how EXPRESSO builds gene expression prediction models using an eQTL dataset that includes both genotype and gene expression levels. We then describe how to analyze datasets with only eQTL summary statistics.

In EXPRESSO, we use epigenomic and 3D genomic information to prioritize genomic regions containing causal eQTL variants, which improves prediction accuracy compared to methods that only consider variants within a contiguous window surrounding each gene. We define essential variants as the ones that overlap at least one of four epigenomic annotation tracks, H3K27ac, H3K4me3, DNase hypersensitivity, or CTCF binding from the ENCODE database. These categories are chosen because of their relevance in gene expression regulation and their broad availability across different tissues and cell types. We denote genotypes of essential variants as $X_e$. We denote variants that

do not overlap the above epigenomic annotation as non-essential variants, i.e., $X_{ne}$. Collectively, we define the genotype matrix as

$$\mathbf{X} = [\mathbf{X_e}, \mathbf{X_{ne}}] \quad (1)$$

and the corresponding effects as

$$\boldsymbol{\beta} = \begin{bmatrix} \boldsymbol{\beta_e} \\ \boldsymbol{\beta_{ne}} \end{bmatrix} \quad (2)$$

Gene expression prediction is based on a linear model,

$$\mathbf{y} = \mathbf{X}\boldsymbol{\beta} + \boldsymbol{\epsilon} = \mathbf{X_e}\boldsymbol{\beta_e} + \mathbf{X_{ne}}\boldsymbol{\beta_{ne}} + \boldsymbol{\epsilon} \quad (3)$$

We fit the model with a hybrid of $L_1$ and $L_2$ penalties, i.e.,

$$
\begin{aligned}
L(\boldsymbol{\beta}; \lambda, \phi, w) &= ||\mathbf{y} - \mathbf{X_e}\boldsymbol{\beta_e} + \mathbf{X_{ne}}\boldsymbol{\beta_{ne}}||_2^2 + \frac{1}{2} \times \frac{\lambda}{2}\left(\phi||\boldsymbol{\beta_e}||_2^2 + ||\boldsymbol{\beta_{ne}}||_2^2\right) \\
&+ \frac{\lambda}{2}\left(\phi||\boldsymbol{\beta_e}||_1^1 + ||\boldsymbol{\beta_{ne}}||_1^1\right) = \mathbf{y^T y} + \boldsymbol{\beta^T X^T X}\boldsymbol{\beta} - 2\boldsymbol{\beta^T X^T y} \\
&+ \frac{1}{2} \times \frac{\lambda}{2}\left(\phi||\boldsymbol{\beta_e}||_2^2 + ||\boldsymbol{\beta_{ne}}||_2^2\right) + \frac{\lambda}{2}\left(\phi||\boldsymbol{\beta_e}||_1^1 + ||\boldsymbol{\beta_{ne}}||_1^1\right)
\end{aligned}
\quad (4)
$$

where $|| \,||_1$ and $|| \,||_2$ denote the $L_1$ and $L_2$ norms. $\lambda$ is the shrinkage parameter, which controls for $L_1$ and $L_2$ penalty. $\phi$ is the mitigation parameter that reduces the shrinkage for essential predictors, so that they are more likely to be retained in the model. We also consider different choices for regions that harbor cis-regulatory variants as another tuning parameter (denoted as $w$), including linear windows surrounding gene start and end sites or regions informed by 3D genome (i.e., loop, TAD, domains, and promoter capture Hi-C (pcHi-C) regions).

It is easy to see that we can calculate the objective function $L(\boldsymbol{\beta}; \lambda, \phi, w)$ with only summary association statistics. For example, we can estimate $\mathbf{X^T X}$ from the LD matrix using a reference panel of matched ancestries and $\mathbf{X^T y}$ is proportional to the marginal eQTL effect size estimates[17]. One challenge in fitting the EXPRESSO model without individual-level data is to estimate tuning parameters. A standard way to estimate tuning parameters is to use cross-validation (CV): in each step, for each set of tuning parameters, a portion of the sample is used to train the model, and the remaining dataset is reserved to validate the accuracy. Overall accuracy is estimated by averaging across all CV folds. The set of tuning parameters that yield the smallest errors is selected. Traditional CV requires individual-level data, but recently, CV methods have been extended to use summary statistics as input[18]. Since CV has to split samples for training and validation, it reduces training sample sizes and hence the accuracy of tuning parameter selection. Here, we also develop a novel approach pseudo-variable selection (PVS) based on simulating unassociated "pseudo-variables" with identical LD structures as measured predictors. We choose the shrinkage parameters that select the largest numbers of measured predictors but none of the pseudo-variables. For further information, please refer to the METHODS and Supplementary Methods sections.

We also implemented another parameter selection strategy that is based on summary statistics-based CV and uses minimal squared error (MSE) as the metric to select tuning parameters (EXPRESSO-MSE). We show in later sections of RESULTS that EXPRESSO-PVS often yields better prediction performance than PUMICE and EXPRESSO-MSE, which optimize the same loss functions.

We conduct extensive simulations to evaluate the proposed methods and compare them with alternative approaches including TWAS methods that use individual-level genotype and expression information, i.e., PUMICE[13], PrediXcan[5], FUSION[4], TIGAR[14], EpiXcan[3], MOSTWAS (DePMA and MeTWAS)[19] and multi-tissue method UTMOST[6], summary statistics based polygenic risk score (PRS) methods PUMAS[18], LDpred2[20], SDPR[21], PRScs[22], LASSOSUM[23], pruning and thresholding (P + T)[24], and two summary statistics based TWAS

methods SUMMIT[25] and OTTERS[26], which adapt four summary statistics based PRS methods, including SDPR, PRScs, LASSOSUM, and P + T.[21–24]. We provide more detailed descriptions in Supplementary Notes, Supplementary Data 1–6, and Supplementary Fig. 1. All p-values in the results are two-sided unless otherwise stated.

## EXPRESSO-PVS substantially improves prediction accuracy when trained using eQTLGen summary statistics

We next evaluate whether large eQTL summary statistics datasets (e.g., eQTLGen) improve prediction accuracy. We use the whole blood dataset from the genotype tissue expression project (GTEx[27] version 7 European sample; $n = 303$) as the training dataset and use Depression Gene Network (DGN[28] European sample; $n = 873$) as testing dataset to assess the prediction accuracy of trained models for all methods. We

noted that eQTLGen includes GTEx whole blood tissue and DGN in the meta-analysis. To ensure the independence between training and test data, we "subtract" the eQTL effects in GTEx and DGN data from eQTLGen according to the formula of inverse variance weighted fixed-effect meta-analysis. We generated one new set of summary statistics, including cohorts other than GTEx and DGN, which we denote as eQTLGen/GTEx/DGN, and used them as training data for methods based on summary-level data (Fig. 1, Supplementary Data 7).

Depending on the training dataset, we compared the prediction accuracy in DGN using four different methods x training dataset combinations, including:

(1) summary statistics-based methods trained on eQTLGen/GTEx/DGN: EXPRESSO-PVS, EXPRESSO-MSE, SUMMIT, OTTERS (which includes pruning and thresholding with p-value cutoffs of 0.001

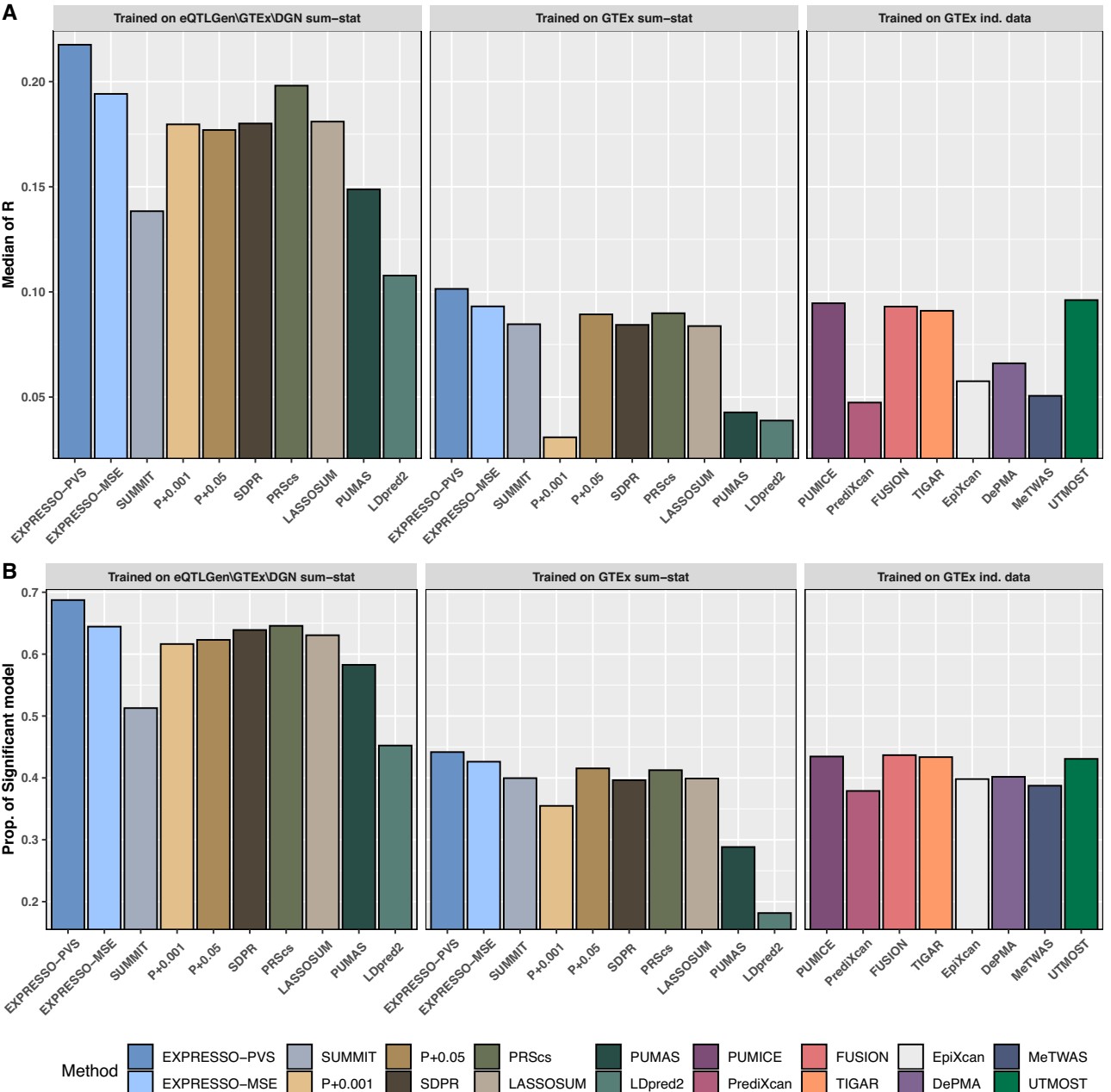

**Fig. 1 | Comparison of gene expression prediction accuracy using DGN as a test dataset.** Panels **A**, **B** compare TWAS methods for (**A**) the median of Pearson's correlation and (**B**) the proportion of significant models. In each comparison, we stratify the methods into three different groups: left columns are methods trained on one QTLGen/GTEx/DGN summary statistics; middle columns are results for methods trained on GTEx summary statistics; right columns are methods trained on individual-level data from GTEx. UTMOST is trained using all 48 tissues, while the other methods are trained using whole blood only.

and 0.05 (P + 0.01 and P + 0.05), SDPR, PRScs, LASSOSUM), PUMAS and LDpred2;

(2) summary statistics-based methods trained on GTEx whole blood summary statistics: EXPRESSO-PVS, EXPRESSO-MSE, SUMMIT, OTTERS (P + 0.001, P + 0.05, SDPR, PRScs, LASSOSUM), PUMAS and LDpred2;

(3) individual-level data-based single-tissue methods trained on GTEx whole blood tissue data: PUMICE, PrediXcan, FUSION, TIGAR, EpiXcan, DePMA and MeTWAS[19]);

(4) individual-level data-based multi-tissue method trained on GTEx data of 48 tissues: UTMOST[6].

EXPRESSO-PVS trained on eQTLGen/GTEx/DGN yields substantially increased numbers of significant models and prediction accuracy, as it analyzes much larger sample sizes than single-tissue and multi-tissue methods that rely on individual-level data. When compared to PUMICE, PrediXcan, FUSION, TIGAR, EpiXcan, DePMA and MeTWAS trained on GTEx whole blood tissue, EXPRESSO-PVS trained on eQTLGen/GTEx/DGN leads to an average increase of 58.17%, 81.41%, 57.37%, 58.54%, 72.65%, 71.09% and 77.44% for the number of significant prediction models (Fig. 1) and also leads to 129.92%, 359.06%, 133.90%, 139.00%, 278.13%, 229.32% and 330.00% increase in the median of Pearson's correlation. Moreover, compared to UTMOST trained on all 48 tissues in GTEx, EXPRESSO-PVS leads to a 59.58% increase in the number of significant models and a 126.33% increase in Pearson's correlation. EXPRESSO-PVS also outperforms the summary statistics-based PRS methods when they are all trained on eQTLGen/GTEx/DGN summary statistics. Compared to EXPRESSO-MSE, SUMMIT, P + 0.001, P + 0.05, SDPR, PRScs, LASSOSUM, PUMAS, and LDpred2, EXPRESSO-PVS increases the Pearson correlation by 12.05%, 57.21%, 21.06%, 22.91%, 20.80%, 9.83%, 20.20%, 46.20%, and 101.84% and increases the number of significant models by 6.66%, 34.04%, 11.52%, 10.33%, 7.57%, 6.45%, 9.01%, 17.97% and 51.98%.

## EXPRESSO substantially improves the prediction accuracy of gene expression at the cell type level

To better understand how the effects of predicted expression vary within and across cell types, we analyze the sc-eQTLGen consortium datasets based on 104 individuals. To utilize these datasets for TWAS, we trained our summary statistics TWAS model on sc-eQTLGen data and used the Database of Immune Cell Expression (DICE[29], n = 91) to evaluate prediction performance. We focus on seven overlapping cell types between sc-eQTLGen and DICE, including unstimulated B cells, unstimulated CD4 T cells, stimulated CD4 T cells, unstimulated monocytes, unstimulated NK cells, unstimulated CD8 T cells, and stimulated CD8 T cells.

EXPRESSO utilizes annotated epigenomic and 3D genomic data to prioritize causal variants. Yet, there is no 3D genomic or epigenomic data for unstimulated NK cells, unstimulated CD8 T cells, and stimulated CD8 T cells. To identify a proxy, we cluster cell types based on expression profiles and eQTL effect sizes (Supplementary Fig. 2). We use 3D genomic data from the nearest neighboring cell types in the clustering analysis as a proxy, as tissues with similar global gene expression profiles tend to have similar 3D genome structures[30]. Based on this analysis, we identify unstimulated B cells, unstimulated CD4 T cells, and stimulated CD4 T cells as proxies for NK cells, unstimulated CD8 T cells, and stimulated CD8 T cells, respectively, and incorporate 3D genomic and epigenomic from these proxy cell types in the analysis.

We apply EXPRESSO-PVS and other summary statistics-based methods for individual cell types. EXPRESSO has better performance than all other summary statistics-based PRS methods (Supplementary Data 8). EXPRESSO-PVS and EXPRESSO-MSE have comparable performance due to the limited sample size of sc-eQTLGen, which is consistent with simulation results. Compared to the second-best method

PRScs, EXPRESSO-PVS increases average prediction accuracy by 14.71% and the proportion of significant models by 10.48%. Interestingly, for the four cell types with corresponding epigenomic and 3D genomic data, we observe a much bigger increase in prediction accuracy (i.e., 18.45%) with EXPRESSO-PVS compared to PRScs. The average improvement is only 11.34% for cell types that lack matched 3D genomic or epigenomic data and have to rely on annotation information from proxy cell types. Our results demonstrate the utility of annotation information for improving prediction accuracy. They underscore the importance of incorporating matched biological information for predicting gene expression levels and the necessity of generating 3D genomic and epigenomic data from diverse cell types.

To further investigate how different annotation tracks influence prediction accuracy, we examine the choice of different window sizes (w) and mitigation factors ($\phi$) among generated EXPRESSO-PVE prediction models (Supplementary Fig. 3). Interestingly, 3D genome-informed regions (loop, TAD, domain, pcHi-C) are chosen 47.27% of the time. On the other hand, the 1 million basepair window, the default window size for many different methods, is only chosen 31.28% of the time. The most frequent choice for mitigation parameter is $\phi = 1/6$ (31.01%), which prioritizes essential predictors by assigning much smaller $L_1$ and $L_2$ penalties. These results demonstrate the utility of using 3D genomic and epigenomic annotations to prioritize causal eQTL variants and improve gene expression prediction accuracy.

## TWAS analysis of 14 autoimmune diseases identifies disease-relevant cell types

We perform TWAS using the gene expression prediction models above trained on whole blood tissue. We integrate the prediction models with multi-ancestry GWAS summary statistics using TESLA[31], a method we developed to optimally integrate eQTL datasets with multi-ancestry GWAS. We analyze 14 autoimmune diseases, including systemic lupus erythematosus (SLE), Crohn's disease (CD), primary biliary cirrhosis (PBC), rheumatoid arthritis (RA), ulcerative colitis (UC), vitiligo (VIT), ankylosing spondylitis (AS), autoimmune thyroid disease (Grave's disease and Hashimoto thyroiditis) (ATD), celiac disease (CELIAC), multiple sclerosis (MS), psoriatic arthritis (PSOA), Sjogren's syndrome (SJO) and type 1 diabetes (T1D), with maximum sample size being 728,548 (Supplementary Data 9 & Supplementary Fig. 4).

EXPRESSO-PVS outperforms the other TWAS methods in terms of the total number of loci, novel loci, and known loci identified compared to the GWAS catalog (Supplementary Data 10&11 and Fig. 2) and yields bigger mean chi-square association statistics at known loci, all of which have been used as metrics to compare different methods. We follow an iterative procedure to define locus: we rank significant genes by TWAS p-values, where smaller p-values are ranked higher. For the top gene, we define a locus as a 1 million basepair window surrounding it. To define the next locus, we follow the same procedure by focusing only on significant genes that are not included in the previously defined loci. We repeat this process until we exhaust the list of significant genes. All methods have well-calibrated genomic control values (Supplementary Data 12). If we assume a majority of the reported loci are genuine, the number of known loci identified and the mean chi-square test statistic (which estimates the non-centrality parameter) at known loci by different methods can be used for empirical power comparison. EXPRESSO-PVS increases the total number of loci by 31.14%, 26.59%, 212.86%, 236.92%, 38.61%, 36.88%, 96.41%, 253.23% and 305.56% compared to other summary statistics-based methods EXPRESSO-MSE, SUMMIT, P + 0.001, P + 0.05, SDPR, PRScs, LASSOSUM, PUMAS and LDpred2 (trained on eQTLGen/GTEx/DGN). It should be noted that OTTERS use Cauchy combination method to combine the p-values of different TWAS methods which yields further improvement of TWAS. EXPRESSO outperformed each method that OTTERS combines. Importantly, adding EXPRESSO to the

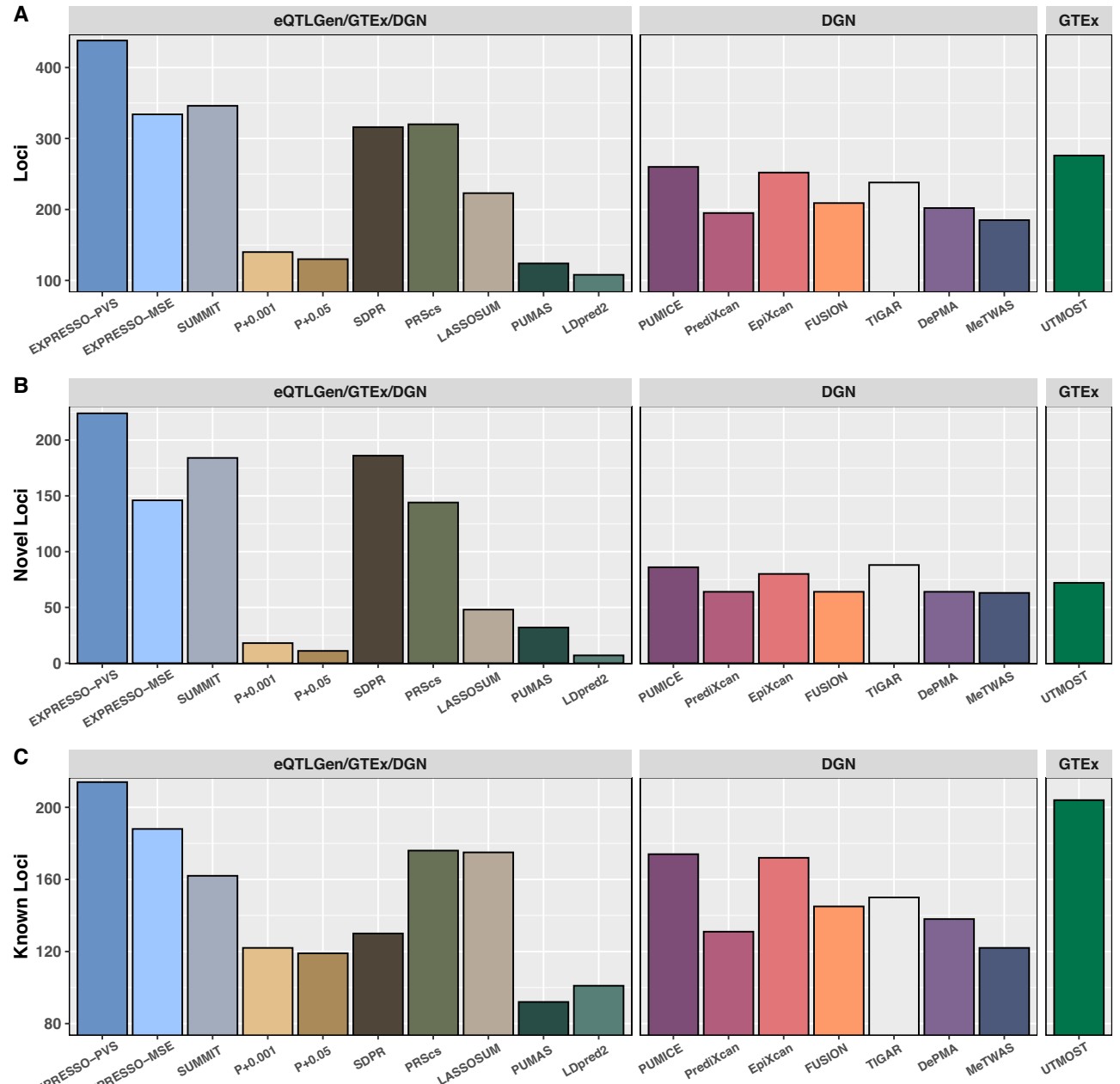

**Fig. 2 | EXPRESSO identifies the largest number of loci in TWAS.** Panels **A**–**C** compare different methods for (**A**) the total number of loci, (**B**) the number of novel loci that do not overlap GWAS catalog and (**C**) the number of known loci that overlap GWAS catalog. We define a locus as novel (or known) if the sentinel variant of the locus is greater than (or within) 1 million base pairs away from reported hits in GWAS catalog. In each comparison, we stratify the methods into three different

groups. In the left panels, we show results for methods trained using eQTLGen/GTEx/DGN summary statistics; in the middle panels, we show results for methods trained using individual-level data from DGN (the largest individual RNASeq data we can access); in the right panels, we show results for methods trained using individual-level data from multiple tissues in GTEx.

set of methods that is combined, we can identify many more loci (686 vs 488), more known loci (259 vs 221) and yield higher mean $\chi^2$ statistics at known loci (36.14 vs 35.84), which all demonstrate the value of EXPRESSO.

When comparing to individual-level data methods PUMICE, PrediXcan, FUSION, TIGAR, EpiXcan, DePMA, and MeTWAS trained on DGN, EXPRESSO-PVS increases total number of loci by 68.46%, 124.62%, 73.81%, 109.57%, 84.03%, 116.83% and 136.76%. EXPRESSO-PVS also found 58.70% more loci than UTMOST trained on GTEx. Finally, we compare the power of different methods using the mean value of chi-square statistic at known loci as a metric, which estimates the non-centrality parameter of chi-square test statistic[32]. EXPRESSO-PVE

consistently yields bigger values of median Z-scores compared to other methods (Supplementary Data 10).

To identify disease-relevant cell types, we performed enrichment analysis to examine cell-type-specific TWAS hits (Supplementary Fig. 5). We identified many disease-relevant cell types with strong biological support. For example, TWAS hits for CD are enriched in classical monocytes and B-cells, both of which are critical in CD etiology. Specifically, *IL-23* induces the secretion of IL-17 by non-T-cells in an inflammatory environment, and both T cells and monocytes serve as sources of increased expression of *IL-23* in the mucosa of IBD patients[33]. Moreover, previous study shows that patients with CD have chronic, aberrant B-cell response. As another example, enrichment

analysis identifies B cells as the most relevant cell type for SLE. SLE is characterized by B cell dysfunction that results in the production of pathogenic autoantibodies. SLE B cells also possess altered antigen presentation and cytokine secretion compared to that in normal individuals[34,35]. To validate the enrichment analysis, we also used neuronal cell types as a negative control, as most of the diseases (except for multiple sclerosis) are not related to brain cell types. As expected, nearly all brain cell types are not enriched with TWAS signals from whole blood and immune cell types. These results using negative controls yield expected false positive rates and help establish the validity of our cell-type enrichment analysis (Supplementary Data 13).

## Cell type level TWAS identifies and fine maps novel loci in disease-relevant cell types

To identify cell-type specific target genes, we perform TWAS using models based on EXPRESSO-PVS and other summary statistics-based methods on seven cell types included in sc-eQTLGen. We also performed fine-mapping to identify causal genes in each cell type.

All methods for cell type level TWAS have well-controlled type I errors (Supplementary Data 12). We define known loci as the 1 million

base pair window surrounding the reported signals in the GWAS catalog. EXPRESSO-PVS increases the total number of loci by 13.36%, 53.26%, 157.99%, 52.41%, 35.81%, 27.01%, 62.97%, 74.36% and 80.09%, increases the number of significant associations by 11.01%, 52.31%, 182.60%, 50.39%, 46.26%, 52.31%, 69.86%, 79.07% and 95.51%, and increases the mean $\chi^2$ statistics at known loci by 17.66%, 36.04%, 132.11%, 15.68%, 29.38%, 29.28%, 27.53%, 24.46% and 30.61% when compared to EXPRESSO-MSE, SUMMIT, P + 0.001, P + 0.05, SDPR, PRScs, LASSOSUM, PUMAS and LDpred2 (Fig. 3, Supplementary Data 14).

Cell-type level TWAS uniquely identifies many GTAs that are missed in whole blood. To understand what drives these cell type only GTAs, we first define "cell type-specific genes" as the ones with expression levels [measured in transcript per million (TPM)] in a given cell type being 1 standard deviation above the mean expression level across all cell types, following the procedure of Boyle et al[36]. We note that cell type only GTAs are more likely driven by cell type specific genes with high expression in that cell type. Rare cell types tend to have larger fractions of cell type specific genes, with a Pearson correlation of −0.75 between cell type proportion and the fraction of cell

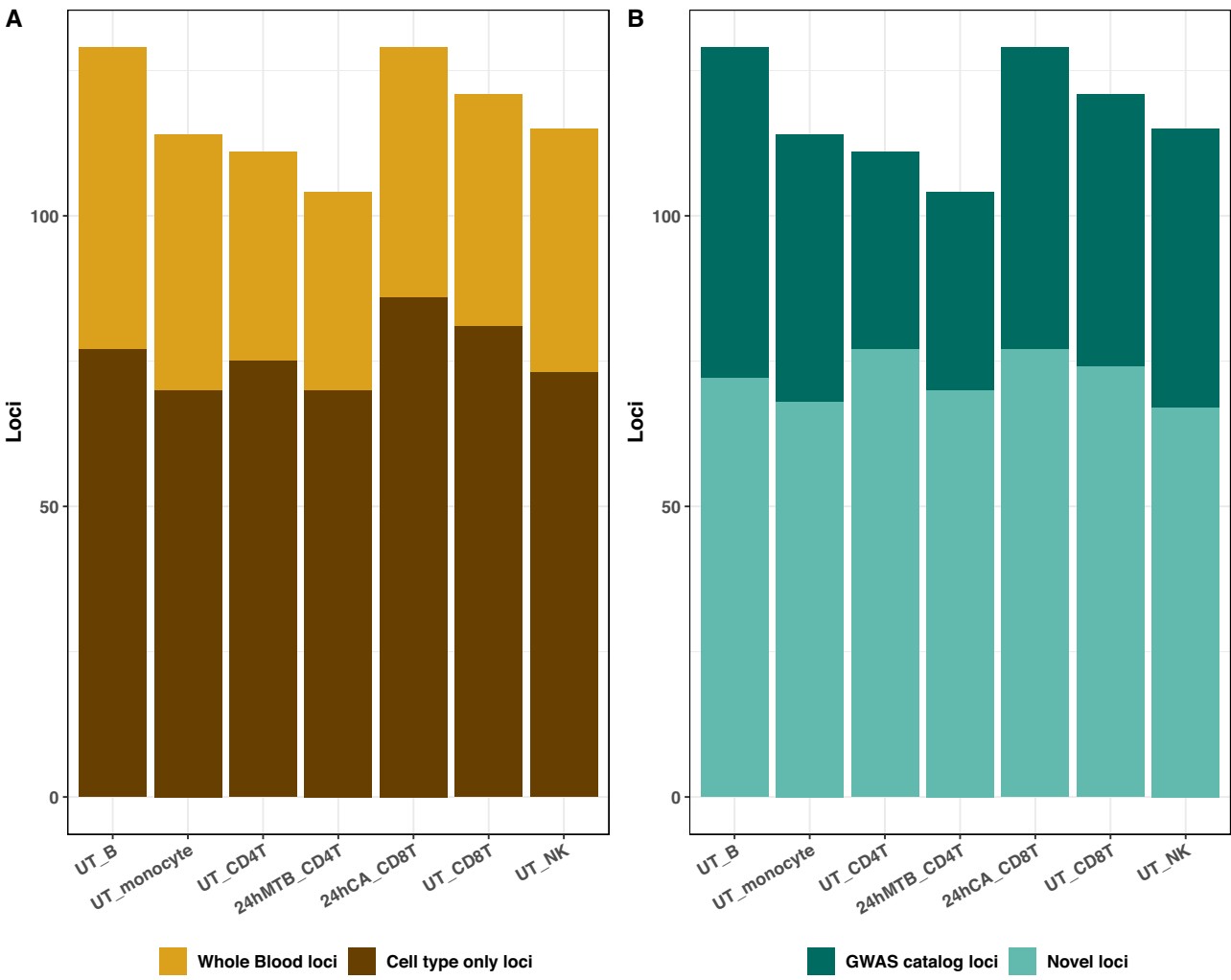

**Fig. 3 | Cell type level TWAS identifies novel loci.** We show the number of loci in seven cell types, including unstimulated B cells (UT_B), unstimulated NK cells (UT_NK), unstimulated CD4 T cells (UT_CD4T), unstimulated CD8 T cells (UT_CD8T), unstimulated natural killer cells (UT_NK), CD4 T cells after 24 hour in vitro stimulation with *M. tuberculosis* (24hMTB_CD4T) and CD8 T cells after 24 hour in vitro stimulation with *C. albicans* (24hCA_CD8T), identified using gene expression prediction models from EXPRESSO. In Panel **A**, we show the number of loci identified in whole blood and the number of loci identified only by cell type

level TWAS (i.e., cell type only loci). In panel **B**, we show the number of novel loci identified in each cell type by cell type level TWAS. We consider a locus as novel if the sentinel variant is greater than 1 million basepair away from reported hits of the same trait in GWAS catalog. We stratify the number of loci by whether they are novel or not. It is clear that cell type level TWAS uniquely identifies many (novel) loci that are missed in whole blood, demonstrating the power of cell type level TWAS.

type specific genes (Supplementary Fig. 6). Additionally, rare cell types tend to harbor more cell type only GTAs that are missed by whole blood TWAS (Supplementary Fig. 6), with a Pearson correlation of −0.42 between cell type proportion and the proportion of cell type only GTAs. We also find that the gene prediction models of cell type only GTAs tend to include a larger number of cell type-specific essential variants than that of whole blood GTAs (Supplementary Data 15). These results together suggest cell type only GTAs are more likely driven by cell type specific risk genes and essential regulatory variants.

Using TESLA, we further fine mapped loci identified by whole blood tissue and seven cell types for 14 autoimmune related traits (Supplementary Data 16). Although whole blood TWAS identifies more loci (438) than the average of each individual cell type (118), we can fine map cell type level TWAS loci with higher resolution. Specifically, we fine map 84.61% of loci in cell type level TWAS to a single gene in the 90%-credible set, but can only do so for 57.75% of whole blood TWAS loci. When we limit our comparison to the loci identified by both cell type level and whole blood TWAS, we still observe improved fine-mapping resolution using cell type level TWAS results (i.e., 89.87% of loci to single gene resolution using cell type level TWAS against 74.42% using whole blood TWAS results). The heterogeneity of predicted gene expression effects across cell types potentially reduces the fine-mapping resolution when whole blood TWAS results are used.

We are able to identify and fine map a number of novel genes which lie > 1 megabase from reported GWAS hits. Many of those novel putatively causal genes have strong biological relevance (Supplementary Data 17) and are uniquely identified in cell type level TWAS but missed in whole blood. One gene identified for rheumatoid arthritis is *C-C Motif Chemokine Receptor 1 (CCR1)* (p-value $= 1.42 \times 10^{-11}$, $PIP = 1$) in unstimulated NK cells. The *CCR1* expression prediction model contains 3 variants, all of which are NK cell-specific essential variants. *CCR1* is involved in the *cellular response to cytokine stimulus* (GO:0071345), *Nociception Expression Targets Signaling*, and *Interleukin-2 signaling pathway* in unstimulated NK cells. Inhibition of *CCR1* improves the symptoms in a mouse model of arthritis[37], which suggests *CCR1* is a potential target for rheumatoid arthritis treatment[38,39].

Interestingly, for SLE, we identify *Platelet Derived Growth Factor Receptor Beta (PDGFRB)* (p-value $= 2.89 \times 10^{-6}$, $PIP = 1$) in naïve B cells. The *PDGFRB* expression prediction model contains 10 causal variants, eight of which are essential variants specific to B cells. PDGF-B resides in *cytokine-cytokine receptor interaction* pathways in naïve B cells. A previous study found the PDGF-B pathway to be excessively activated in SLE patients[40]. The PDGFRA receptor has been associated with SLE[41] and our study now links *PDGFRB* to SLE as well.

Perez et al.[10] also conducted cell type-specific TWAS using CONTENT, which decomposes eQTL effects into shared and cell type-specific components. The study pinpointed risk genes associated with SLE and other autoimmune-related conditions, including CD and RA. To compare with their results, we focus on novel loci in five cell types that are commonly measured between the two studies, i.e., B cells, classical monocytes, natural killer cells, CD4 T cells, and CD8 T cells. Our study uniquely identified 98 genes that are not in Perez et al., including 60 novel genes beyond the 1 million base pair window of GWAS catalog-identified loci. We also identified 54 novel genes for RA and 97 novel genes associated with CD.

## Characterizing gene expression effect heterogeneity across cell types

We first examined overlaps between TWAS signals identified in whole blood and by cell type level analysis. Among the 1222 GTAs identified in either whole blood or in individual cell types, ~50% (626) are only identified at cell-type level analysis. The overlap between whole blood and cell type level TWAS hits is ~10% (115). The low overlap between whole blood and cell type level TWAS may reflect phenotypic effect

heterogeneity between cell types or may arise simply because of limited power for detecting associations in whole blood or individual cell types.

To assess the heterogeneity of TWAS effect sizes more rigorously, we develop a new statistical method to compare TWAS effect sizes across cell types (**METHODS**). Among the 1222 genes that are significant in either whole blood or cell type level TWAS analysis, 31.2% of tested genes show statistically significant differences in TWAS effects between cell types (p < 0.05/1222 = $4.1 \times 10^{-5}$). The fraction of genes with significant TWAS effect heterogeneity increases to 37.06% if we restrict the analysis to the genes identified only in cell type level TWAS.

We further visually examine effect heterogeneity across cell types. We plot effects across different cell types and whole blood for genes with significant p-values in at least one cell type (Fig. 4, Supplementary Fig. 7, and Supplementary Data 18). As shown, for many genes that are identified only in cell type level TWAS, the TWAS effect size heterogeneity is often large. On the other hand, when TWAS effect sizes are homogeneous between cell types, whole-blood TWAS statistics often have more significant p-values due to the large sample sizes of the whole-blood eQTL dataset.

Our results corroborate previous studies on cell type specificity of eQTL effects[8] and further characterize how the phenotypic effects of predicted gene expression vary between cell types. TWAS analysis using bulk RNASeq datasets may frequently miss genes with significant heterogeneity across cell types, thus underscoring the importance of conducting functional genomic analysis at the cell type level.

## Cell type aware computational drug repurposing using TWAS based on sc-RNASeq data

We developed a Cell type Aware Drug REpurposing pipeline (CADRE) to perform computational drug repurposing using TWAS results from both single-cell and bulk-RNASeq data. We use enrichment analysis to pinpoint disease-relevant cell types and identify cell lines that closely mimic the transcriptomic profile of disease-relevant cell types (Supplementary Fig. 3 and Supplementary Data 13&19). We then use the CMap database[42] and TWAS results based on sc-RNASeq to identify small bioactive molecules capable of reversing the expression profile of clinically relevant trait-associated genes, which could be repurposed to treat autoimmune diseases. To link disease to drug-induced states, CMap computes a $\tau$ score using the cell line that closely resembles disease-relevant cell types (Supplementary Fig. 8) to evaluate both query features and references. A negative $\tau$ score indicates that the molecule will normalize the gene expression profile associated with the trait and can potentially be repurposed to treat disease.

We compare results from our cell-type CADRE pipeline to those based on the whole blood RNASeq dataset (Fig. 5 and Supplementary Data 20). Drugs implicated by our CADRE analyses yield much lower average $\tau$ scores (−5.49) than those obtained using whole blood TWAS results (−2.64), indicating that CADRE can identify drugs that more consistently reverse disease gene expression levels compared to drug repurposing analysis conducted using whole blood samples.

We also seek to demonstrate the utility of cell type specific TWAS results in drug repurposing using an orthogonal approach based on drug target gene enrichment analysis. We first generate gene sets for drug targeted pathways for each drug using DrugBank[43], a database that documents the mechanism of action for all approved drugs. We then perform drug target enrichment analysis based on TWAS results. We compare the enrichment p-value between repurposing analysis using TWAS results from disease relevant cell types and that using whole blood. Among the drugs either approved for treatment or for clinical trial, 12 show nominally significant enrichment with cell type specific TWAS hits from disease relevant cell types, and only 2 show enrichment with TWAS hits from whole blood, demonstrating significant differences (with two-sided Fisher's exact p-value 0.008) (Supplementary Data 20). Besides, 88.9% of the drugs are more

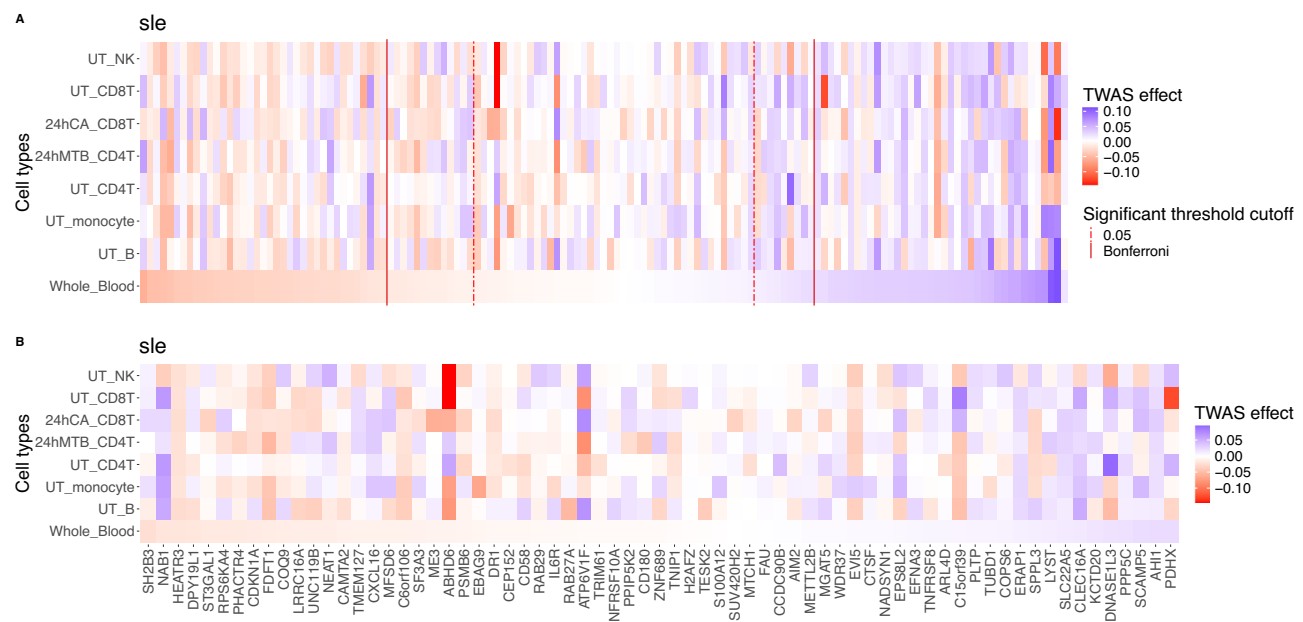

**Fig. 4 | Phenotypic effects of predicted gene expression across cell types for SLE.** Genes are ordered by the TWAS Z-scores in whole blood. Panel **A** shows the TWAS effects across different cell types and whole blood for each gene. Panel **B** focuses on gene x trait associations identified only in cell type level analysis, which zooms in the middle part of Panel **A**, between the red solid line. The genes in panel **B** have two-sided whole blood TWAS Bonferroni-corrected *p*-values > 0.05 (Red dashed line represent significant threshold cutoff under two-sided *p*-value = 0.05, red solid line line represent significant threshold cutoff under Bonferroni correction with two-sided *p*-value = 0.05/number of genes (1500)). The results show that genes with heterogeneity effects across cell types are often missed in whole blood TWAS. The phenotypic effects of predicted gene expressions for other autoimmune traits are included in Supplementary Fig. 7.

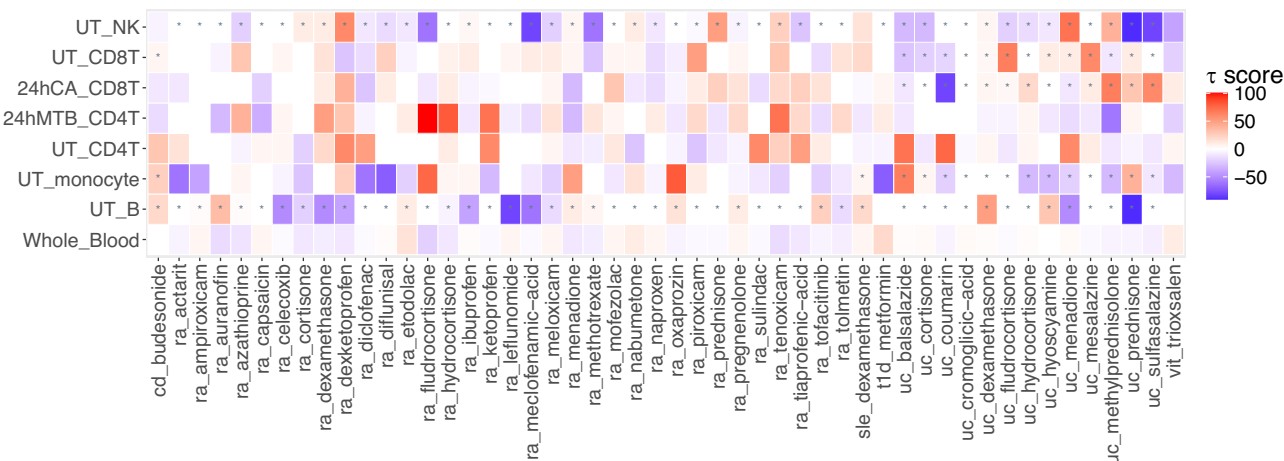

**Fig. 5 | Distribution of τ scores from computational drug repurposing.** In CADRE, we pinpoint disease relevant cell types using enrichment analysis. We then use CMap cell lines with matched transcriptome profiles to identify drugs that may reverse disease gene expression. The figure shows the τ scores for different cell types and whole blood tissue. We denote significantly enriched cell type with * (two-sided *p* < 0.05/15, which is the Bonferroni threshold for testing 15 cell types).

strongly enriched with cell type specific TWAS hits from disease relevant cell types than whole blood. These multiple lines of evidence further establish the advantage of our cell type aware drug repurposing pipeline.

CADRE analyses identify a larger number of drug classes that show promise in the clinic than using whole blood data alone, and those drugs were usually closely related to the enriched cell types. We identified cyclooxygenase 2 (COX-2) inhibitors, dihydrofolate reductase inhibitors, and glucocorticoid receptor agonists for RA, for which B cells and NK cells are the most enriched cell types with TWAS hits. Of note, these drugs have already been used to treat RA in the clinic[44–46]. The effects of these drugs in B and NK cells are well established, which helps confirm the validity of this approach[44,45] and explain cell type enrichment effects. Specifically, methotrexate, a dihydrofolate reductase inhibitor, decreases the number of transitional B cells and serum immunoglobulin levels in arthritis patients[47]. We also identify a glucocorticoid receptor agonist, prednisone, which has been used for treating RA[46]. Prednisone in low doses can suppress the inflammation associated with RA[48], restrict B lymphocyte differentiation into plasma cells[49], and suppress the cytolytic activity of NK cells[50].

CADRE also identifies novel drug classes with biological and medical relevance, including metformin for T1D and vitamin K for UC. Metformin, an insulin sensitizer, has been widely used in conjunction with diet and exercise for glycemic control in type 2 diabetes mellitus patients. Recent studies suggest that adding metformin to pharmacologic insulin dosing in type 1 diabetics may be effective, as

metformin can decrease glucose concentrations and reduce metabolic syndrome[51]. Additionally, menadione, also known as vitamin K3, was identified for UC, where CD4 T and CD8 T cells are the two cell types most enriched with TWAS signals. The role of vitamin K on intestinal health has drawn growing interest in recent years. Studies have shown that vitamin K presents a beneficial effect on intestinal health[52], which can affect immune and inflammatory responses mediated by T cells[53]. Our study now further supports vitamin K for UC.

### Computational efficiency

We provide a documented and open-source implementation of EXPRESSO (see Code Availability). The software is efficient and takes <24 hours to analyze the whole genome on a standard server with Intel(R) Xeon(R) CPU E5-2680 v3 @ 2.50 GHz, 32 GB RAM, and a 7200 rpm hard drive.

## Discussion

In this article, we present a new method EXPRESSO (Supplementary Fig. 9) to integrate eQTL summary statistics from sc-RNASeq and bulk-RNASeq to identify target genes associated with complex traits/ diseases. Importantly, EXPRESSO incorporates a novel method pseudo-variable selection to more effectively select tuning parameters without external validation data, addressing an important challenge of analyzing eQTL summary statistics. As the largest eQTL datasets are often available or most conveniently accessible as summary statistics, EXPRESSO allows the analysis of much larger eQTL datasets. It can also integrate 3D genomic and epigenomic information to prioritize causal eQTLs and further improve gene expression accuracy. EXPRESSO outperforms all existing TWAS methods that rely on individual level data. It also substantially outperforms other summary statistics-based prediction methods adapted to predict gene expression.

We apply EXPRESSO to analyze large multi-ancestry GWAS datasets of 14 autoimmune diseases. We identified causal genes that are specific to certain cell types. Using a novel heterogeneity test statistic, we rigorously characterize how the mediating effects of genetically regulated gene expression vary across cell types. It differs from earlier attempts that seek to overlap eQTLs from different cell types, which tend to underestimate the extent of shared effects. Our results show that ~31% of genes show significant differences in TWAS effects between cell types. This is in stark contrast to the overlap analysis showing only 10% of the genes are identified in both whole blood and cell type level TWAS, which is likely an underestimate.

As the effects of predicted expression levels for many genes can be quite different across cell types, these genes are likely missed in TWAS analysis of whole blood. On the other hand, EXPRESSO based on sc-eQTL data reveals numerous causal genes that are cell type dependent. Our drug repurposing pipeline CADRE based on cell type level TWAS identifies drugs that can more consistently reverse gene expression in disease relevant cell types. The drug target pathways of approved drugs are more strongly enriched with TWAS hits from disease relevant cell types than TWAS hits from bulk tissue. Many of the identified new drugs also have support from clinical trials or animal studies and show promise for treating autoimmune diseases, further supporting this strategy.

EXPRESSO relies on 3D genomic and epigenomic data from matched cell types to prioritize causal variants and improve prediction accuracy. For cell types without these data, we cluster transcriptomic profiles from different cell types, identify nearest neighbors as proxies, and use the 3D genomic and epigenomic data from proxy cell types to fit the model. Although we still observe improved prediction accuracy for cell types using proxy annotation information, the improvement is smaller, likely suggesting that some cell type differences are not fully captured with this approach. As single cell RNASeq datasets continue to grow larger, it will be important to generate companion 3D genomic and epigenomic datasets for relevant cell types to better annotate

functional variants, which will also help improve EXPRESSO and similar methods thereof for integrative analysis.

EXPRESSO trains gene expression prediction models for each tissue separately. A few multi-tissue TWAS methods exist that leverage shared regulatory variants across multiple tissues to improve TWAS for tissue types with smaller sample sizes[54,55]. Yet, existing multi-tissue TWAS methods tend to prioritize the selection of causal variants shared between tissues. They often do not perform well for genes with tissue specific eQTLs. We can explore similar ideas for cell type level TWAS. Given the extensive heterogeneity of regulatory variant effects between cell types, it remains unclear if multi-cell type extensions of TWAS methods can work well.

EXPRESSO only considers cis-eQTL effects and does not model trans-eQTL effects. Detecting and modeling trans-eQTL effects require larger sample sizes, which are currently not available. Indeed, studies investigating trans-eQTL effects using sc-RNASeq data reveal very sparse signals. Yet, other studies have shown that for bulk tissue TWAS, modeling trans-eQTL effects may further improve prediction accuracy. Given the advantage of EXPRESSO for modeling expression levels using cis-eQTLs, it is reasonable to expect that extending EXPRESSO to incorporate trans-eQTLs may further improve prediction accuracy.

In conclusion, we present an integrative framework to perform gene-based association analysis. EXPRESSO is built on publicly available eQTL summary data of single cell and bulk-RNASeq, allowing the framework to generate prediction models with higher imputation accuracy and discover cell type-specific risk genes. As the research community continues to generate and assemble large sc-eQTL datasets, EXPRESSO and its future extensions will be valuable for integrative analysis and play an important role for understanding the phenotypic impact of regulatory variants.

## Methods

### Estimate shrinkage tuning parameters using pseudo variable selection (PVS)

To predict gene expression levels, EXPRESSO model seeks to minimize the following loss function, i.e.,

$$L(\beta; \lambda_1, \lambda_2) = ||\mathbf{Y} - \mathbf{X}_{ne}\boldsymbol{\beta}_{ne} - \mathbf{X}_e\boldsymbol{\beta}_e||_2^2 + \frac{1}{2} \times \frac{\lambda}{2}\left(\phi||\boldsymbol{\beta}_e||_2^2 + ||\boldsymbol{\beta}_{ne}||_2^2\right) + \frac{\lambda}{2}\left(\phi||\boldsymbol{\beta}_e||_1^1 + ||\boldsymbol{\beta}_{ne}||_1^1\right) \quad (5)$$

The tuning parameters include the mitigation factor $\phi$ and the shrinkage parameter $\lambda$. While not explicit in the formula, we consider the window sizes (denoted by $w$) as a tuning parameter, which includes linear distance-based windows and 3D genome-based windows. For a given set of tuning parameters, we can estimate the regression parameters of the prediction model, i.e., $\boldsymbol{\beta}_{ne}$ and $\boldsymbol{\beta}_e$ using a modified cyclic coordinate descent algorithm (Supplementary Text).

Among the tuning parameters, the shrinkage parameter $\lambda$ is most critical for prediction accuracy. Here, we propose a new method PVS to select the shrinkage parameter more effectively. PVS generates a set of pseudo variables $\mathbf{X}_\pi$ that have the same covariance structure as the observed set of predictors but are not associated with the phenotypes of interest. Specifically, we introduce an auxiliary loss function that includes both the measured predictors as well as pseudo variables, i.e.,

$$L^*(\boldsymbol{\beta}, \boldsymbol{\beta}_\pi; \lambda, w, \phi) = ||\mathbf{Y} - \mathbf{X}\boldsymbol{\beta} - \mathbf{X}_\pi\boldsymbol{\beta}_\pi||_2^2 + + \frac{1}{2} \times \frac{\lambda}{2}\left(\phi||\boldsymbol{\beta}_e||_2^2 + ||\boldsymbol{\beta}_{ne}||_2^2 + ||\boldsymbol{\beta}_\pi||_2^2\right) + \frac{\lambda}{2}\left(\phi||\boldsymbol{\beta}_e||_1^1 + ||\boldsymbol{\beta}_{ne}||_1^1 + ||\boldsymbol{\beta}_\pi||_1^1\right) \quad (6)$$

Bigger values of $\lambda$ impose stronger penalty on the parameters $\boldsymbol{\beta}$ and $\boldsymbol{\beta}_\pi$ which usually eliminates more variables. Since the pseudo values are not associated with the outcomes, we seek to choose a

shrinkage parameter $\lambda$ to ensure that the model eliminates all pseudo variables and at the same time, retains as many measured variables as possible. Based on this intuition, for each pair of window size and mitigation factor values, we gradually increase the tuning parameter $\lambda$ until all coefficients of the pseudo variables become zero and we denote the resulting tuning parameter as $\hat{\lambda}(w,\phi)$, i.e.,

$$\hat{\lambda}(w,\phi) = \min\left\{\lambda : \hat{\boldsymbol{\beta}}_{\boldsymbol{\pi}}(\lambda,w,\phi) = \mathbf{0}\right\} \tag{7}$$

$\hat{\boldsymbol{\beta}}_{\boldsymbol{\pi}}(\lambda,w,\phi)$ is the solution that minimizes the loss function with tuning parameters $\lambda,w,$ and $\phi$, i.e.,

$$\hat{\boldsymbol{\beta}}(\lambda,w,\phi),\hat{\boldsymbol{\beta}}_{\boldsymbol{\pi}}(\lambda,w,\phi) = \operatorname{argmin}_{\boldsymbol{\beta},\boldsymbol{\beta}_{\boldsymbol{\pi}}} L^{*}(\boldsymbol{\beta},\boldsymbol{\beta}_{\boldsymbol{\pi}};\lambda,w,\phi) \tag{8}$$

When individual data is available, the most straightforward approach to generate pseudo variable is permutation. In the absence of individual level data, we devise a new approach to generate summary association statistics of pseudo variables using Monte Carlo simulation. We note that the covariance between measured predictors, pseudo variables, and gene expression satisfies

$$\mathbf{X}_{\boldsymbol{\pi}}^{\mathbf{T}}\mathbf{X}_{\boldsymbol{\pi}} = \mathbf{X}^{\mathbf{T}}\mathbf{X} \tag{9}$$

$$E[\mathbf{X},\mathbf{X}_{\boldsymbol{\pi}}]^{\mathbf{T}}[\mathbf{X},\mathbf{X}_{\boldsymbol{\pi}}] = \begin{bmatrix} \boldsymbol{\Sigma} & 0 \\ 0 & \boldsymbol{\Sigma} \end{bmatrix} \tag{10}$$

$$E\left(\mathbf{X}_{\boldsymbol{\pi}}^{\mathbf{T}}\mathbf{Y}\right) = 0 \tag{11}$$

where $\boldsymbol{\Sigma} = \mathbf{X}^{\mathbf{T}}\mathbf{X}$ and can be estimated from a reference panel of matched ancestry.

We can simulate the summary statistics for pseudo variables as

$$\mathbf{X}_{\boldsymbol{\pi}}^{\mathbf{T}}\mathbf{Y} \sim N(\mathbf{0},\operatorname{var}(Y) \times \boldsymbol{\Sigma}) \tag{12}$$

In our implementation, we simulate ten sets of pseudo variables. For each set, we estimate tuning parameters $\lambda(\phi,w)$, which we will then average across ten simulated datasets to get final estimates to improve the stability of results.

To select mitigation and window size tuning parameters, we perform cross validation by modifying a recently proposed summary statistics-based CV[18]. We can view the summary statistics in the loss function, i.e., $\boldsymbol{X}^{T}\boldsymbol{y}$, as the sum of genotype-expression covariances from individual study participants, i.e.,

$$\mathbf{X}^{\mathbf{T}}\mathbf{y} = \sum_{i}\mathbf{X}_{\mathbf{i}}y_{i} \tag{13}$$

Where $\mathbf{X}_{\mathbf{i}} = \left(X_{i1},\ldots,X_{ip}\right)$. $\mathbf{X}_{\mathbf{i}\cdot}y_{i} = \left(X_{i1}y_{i},\ldots,X_{ip}y_{i}\right)$ follows a multivariate normal distribution, i.e.,

$$\mathbf{X}_{\mathbf{i}\cdot}y_{i} \sim N\left(\mathbf{X}^{\mathbf{T}}\mathbf{y}/N,\boldsymbol{\Sigma}\right) \tag{14}$$

$\boldsymbol{\Sigma}$ is the estimated variance matrix between genetic variants from a reference panel.

To perform CV, we will simulate $\mathbf{X}_{\mathbf{1}}y_{1}, \ldots, \mathbf{X}_{\mathbf{N-1}}y_{N-1}$ according to the above distribution, and calculate $\mathbf{X}_{\mathbf{N}}y_{N} = \mathbf{X}^{\mathbf{T}}\mathbf{y} - \sum_{i=1}^{N-1}\mathbf{X}_{\mathbf{i}}y_{i}$. To mimic 5-fold CV, we partition the indices of sample individuals into five folds. In each iteration of the CV, we calculate summary statistics for 4/5 of the data to train the model and retain the remaining 1/5 of the data for validation.

For each pair of parameter values of $\phi$ and $w$ and the estimated shrinkage parameter $\hat{\lambda}(\phi,w)$, we evaluate the loss function $L^{*}(\hat{\boldsymbol{\beta}}(\hat{\lambda}(w,\phi),w,\phi),\hat{\boldsymbol{\beta}}_{\boldsymbol{\pi}}(\hat{\lambda}(w,\phi),w,\phi);\hat{\lambda}(w,\phi),w,\phi)$. The tuning parameter

values that yield the minimal loss will be selected and resulting estimates of $\hat{\boldsymbol{\beta}}$ will be used to predict gene expression levels.

## Testing the heterogeneity of TWAS statistics across cell types

In a TWAS, we regress phenotype (residuals) over predicted expression levels of a given gene. We call the regression slope the effect of predicted gene expressions, or simply TWAS effects. We propose a new statistic to rigorously assess the heterogeneity of TWAS effects across cell types. Specifically, we first define the vector of TWAS effects of gene $m$ in different cell types as $\boldsymbol{U_m}$. To simplify notations, we assume a total of $P$ SNPs were used in the prediction of expression in at least one cell type. We encode the prediction weights for each cell type as a vector of length $P$. Some elements in the vector can be zero if the corresponding SNP is not used in the prediction model. We define the LD matrix between the $P$ SNPs as $\boldsymbol{\Sigma}$. For two tissue types $k_1$ and $k_2$, we denote the weights as $\hat{\boldsymbol{\beta}}_{\mathbf{k_1}}$ and $\hat{\boldsymbol{\beta}}_{\mathbf{k_2}}$. The effect estimates of predicted gene expressions across different cell types are correlated. The covariance between two TWAS effect estimates is equal to.

$$\phi_{t_1 t_2} = \hat{\boldsymbol{\beta}}_{\mathbf{k_1}}^{\mathbf{T}}\boldsymbol{\Sigma}\hat{\boldsymbol{\beta}}_{\mathbf{k_2}} \tag{15}$$

We denote the covariance matrix for $\boldsymbol{U_m}$ as $\boldsymbol{\Phi}$. Our null hypothesis is

$$H_0 : E(U_{1m}) = E(U_{2i}) = \ldots = E(U_{Km}) = E(\bar{U}_m) \tag{16}$$

where $\bar{U}_m = \frac{1}{K}\sum_{k=1}^{K}U_{km}$. We could calculate the heterogeneity statistics for gene $m$ as follows:

$$H_m = \sum_{k=1}^{K}\left(U_{km} - \bar{U}_m\right)^2 = \mathbf{U_m^T C U_m} \tag{17}$$

Where $\boldsymbol{C}$ is the centering matrix, i.e., $\boldsymbol{C} = \mathbf{I} - \frac{1}{M}\boldsymbol{11^T}$. It is easy to show that the mean centered TWAS effects satisfy:

$$\mathbf{U_m} - \bar{U}_m \sim MVN\left(0,\mathbf{C\Phi C'}\right) \tag{18}$$

The heterogeneity statistic $H_m$ is a quadratic function of multivariate normal random variables $\boldsymbol{U_m}$. $H_m$ follows a weighted sum of chi-square distributions, i.e., $\vec{T}_m - \bar{T}_m \sim \sum_{k=1}^{K}\lambda_k\chi_{df=1}^2$, with weights $\lambda_k$'s being the eigenvalues for the matrix $\mathbf{C\Phi C'}$.

## Gene x trait association analysis with TESLA

The EXPRESSO prediction model is based on samples of European ancestry. Yet, the GWAS datasets of autoimmune diseases are multi-ancestral. We apply TESLA[31] (Trans-Ancestry Integrative Study using an optimal Linear combination of Association statistics) to conduct TWAS, integrating the European eQTL dataset with GWAS datasets from multiple ancestries. By exploiting shared phenotypic effects between ancestries and accommodating potential effect heterogeneities, TESLA improves power over the TWAS methods using ancestry-matched GWAS and eQTL data and the TWAS methods based on fixed-effect meta-analysis results.

## Cell type enrichment analysis

We retrieved cell type expression from the Database of Immune Cell Expression (DICE). DICE profiled transcriptomic data of 15 immune cell types (2 of which are activated cell types) and included genotype data from 106 samples. We processed the dataset following the pipeline outlined in our previous study[13]. First, we quantify the expression level of each gene using TPM. Next, we compute the average expression for each gene in each cell type. We remove genes not expressed across all cell types. We then rescale gene expression to 1 million TPM for each cell type, to minimize the impact of library size. For each gene, we

define the "gene expression specificity score" by dividing the expression of each gene in a given cell type by the total expression of the same gene across all cell types. We define the cell type-specific genes as the ones with gene expression specificity score in the top 10th percentile in each cell type. We then follow a previously established weighted regression framework to assess if the cell type-specific genes are enriched with significant TWAS hits. We also use neuronal cell types as a negative control, as none of the diseases, with the exception of multiple sclerosis, are related to brain cell types. We performed the same procedures using brain cell type expression data[56]. As expected, nearly all brain cell types are not enriched with TWAS signals from the brain.

**Cell type aware computational drug repurposing pipeline and enrichment analysis**

We develop a cell type aware drug repurposing pipeline (CADRE). CADRE first leverages cell type enrichment analysis to identify disease-relevant cell types. We then cluster the cell lines used in the CMap database and identify cell lines whose transcriptome profiles most closely resemble disease-relevant cell types.

CADRE compares TWAS signatures with the drug perturbation results in CMap database to identify drugs that we may repurpose for treating the disease of interest. Specifically, to identify TWAS signatures, we only choose genes with significant TESLA p-values after the correction of testing multiple genes using the Bonferroni threshold. We restrict our drug repurposing analysis to six traits with known drug indications and having at least 10 positively and 10 negatively associated genes as recommended. We compared TWAS signatures with gene expression changes caused by perturbations in the CMap database (from L1000). We focus only on CMap Touchstone dataset, which contains gene expression patterns from nine cell lines treated with ~3000 well-annotated small-molecule drugs. To link disease to drug-induced states, CMap computes a $\tau$ score using the cell line that closely resembles disease-relevant cell types to evaluate both query features and references. A more negative $\tau$ score indicates that the molecule will more consistently normalize the gene expression profile associated with the trait and can be repurposed to treat disease.

We also perform drug target enrichment analysis as a complementary approach for drug repurposing analysis. For drugs that may be repurposed for treating a disorder, their drug target pathways may be enriched with TWAS signals. By examining the significance of enrichment, we can identify putative drugs. This approach is orthogonal to CADRE. By comparing CADRE results with drug target enrichment results, we can validate identified putative drugs. An improved pipeline would be expected to yield stronger enrichment results. To perform drug target enrichment analysis, we first generate gene sets for drug-targeted pathways for each drug using DrugBank, a database that documents the mechanism of action for all approved drugs. We then perform drug target enrichment analysis using eTESLA, a published multiple regression-based method based on TESLA p-values[31].

**Reporting summary**

Further information on research design is available in the Nature Portfolio Reporting Summary linked to this article.

## Data availability

The pre-trained eQTLGen and sc-eQTLGen gene expression prediction models (hg19 reference genome) generated in this study can be found at https://github.com/LidaWangPSU/EXPRESSO/tree/main/trained_model. The eQTLGen summary statistics are publicly available from https://eqtlgen.org/cis-eqtls.html. The sc-eQTLGen summary statistics are available from https://eqtlgen.org/sc/datasets/1m-scbloodnl-eqtls.html. GTEx V7 data can be

obtained from dbGaP study accession phs000424.v7.p2. DGN data can be requested at https://www.nimhgenetics.org/request-access/how-to-request-access under "Depression Genes and Networks study (D. Levinson, PI)". DICE dataset can be requested through dbGaP accession number phs001703.v1.p1. Epigenomic data were obtained from http://screen.encodeproject.org. 3D genomic data were obtained from http://3dgenome.org. Cell type aware computational drug repurposing analysis was conducted on CLUE Drug Repurposing Hub, which can be accessed at https://clue.io/repurposing-app. GWAS summary statistics files are publicly available, and PubMed ID for each study is provided in Supplementary Data 9. All data supporting the findings described in this manuscript are available in the article and its Supplementary Information files.

## Code availability

Software implementing the EXPRESSO model is available at https://github.com/LidaWangPSU/EXPRESSO.

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

## Acknowledgements
This work was supported by the National Institutes of Health grants R01ES036042, R01IA174108, R01HG011035 to D.J.L and by the Artificial Intelligence and Biomedical Informatics pilot funds from the Penn State College of Medicine. We would like to thank Ms. Qile Dai for help with OTTERS methods and Mr. Zichen Zhang for help with SUMMIT methods and implementation.

## Author contributions
L.W., C.K., L.C., B.J. and D.J.L. conceived the study and developed the statistical model. L.W. and C.K. led the data analysis. L.W., C.K., H.M., D.C. and F.Z. conducted analyses. C.K., H.M. and F.C. helped with interpretation. X.Z. helped with coding and programming. L.W. and D.J.L. prepared the manuscript. All authors contributed to manuscript editing and approved the manuscript. B.J., D.J.L. and L.C. jointly supervised the project.

## Competing interests
The authors declare no competing interests.
