## [Peer Review File · Nature Communications]

Integrating single cell expression quantitative trait loci summary statistics to understand complex trait risk genesReviewer #1 (Remarks to the Author):

Wang et al developed a new method to use summary statistics from eQTL mapping studies to build expression models. The unique feature is that it can leverage epigenomic information. The paper further developed a cell type aware drug repurposing pipeline. For the TWAS prediction model building method, I have some concerns about missing details, the evaluation of this method and the contribution of the method on real data analysis.

My major concern is that the study did not clearly differentiate from current methods that use summary statistics data to build prediction models. Using summary statistics to build prediction models for TWAS have been reported before (SUMMIT Nature communications 2022, OTTER Nature communication 2023), but they are not able to incorporate epigenetic information. Thus, I think the main advancement of the method is being able to use epigenetic/3D-genome information on top of using summary statistics to build the model, however, this has not been described clearly in the manuscript. Instead, most analysis results presented in the paper emphasized on the ability of using summary statistics, e.g allows one to use much larger sample size, and cell type specific models are informative than tissue-level models. This is not related to the major advancement claimed by the paper (include epigenetic/3D genome in model) but solely being able to use summary statistics.

More specifically

Lacking information about the choice of epigenetic annotation. The authors used four annotations tracks from ENCODE. Is this a universal choice? Why these four annotations are chosen and how does different choices of annotations affect the performance of the method?

The author mentioned Hi-C is also used, is this used as a separate class of essential variants? The authors indicated there is another tuning parameter w for Hi-C defined regions. How is it incorporated into the model? The given likelihood function does not have w . Similar to the questions on epigenetic data defined essential variants, how the choice of loop, TAD, and promoter regions affect results? Do the authors suggest to use all these categories to annotate or one category is more useful than the others?

I think being able to account for functional annotation is a major contribution, but how much this information help to improve power, prediction accuracy etc is not clearly shown.

The authors uses a polygenic model in simulations and have two categories of variants with different effect size distributions. I think the current observation is that most genes have a few eQTLs and therefore sparse models should be included in simulations. And how robust it is to model misspecification, e.g. when there are more than two categories of variants effects size distribution?

The authors says "It is clear from this comparison that EXPRESSO already outperforms methods based on individual level data, when they all analyze the same set of individuals." Does this increase in power comes from using epigenetic data? Compared with individual level methods EpiXcan and MOSTWAS (Bhattacharya et al 2021 Plos Genetics) which can also take multiomics data, where does this increase in power comes from?

There are also PRS methods that accounts for annotations, how does the proposed method compared to those? Like in the above comments, I think comparing with methods that can also account for functional annotation is a more direct comparison to help the readers understand the advancement in the methodology.

Results for cell type specific TWAS is clearly showing that cell type specific GWAS is adding useful information to tissue level TWAS. However, I don't think this is directly related to how this proposed method brings new biological knowledge. Cell type specific TWAS has been performed (e.g <https://www.ncbi.nlm.nih.gov/pmc/articles/PMC9297655/>) so I think if the authors show that there

are novel findings by comparing with results from existing analysis and these novel findings makes biological sense, then that would be a more direct evidence of the value of the method.

Other issues:

In general, I feel the the simulation results lack clear description. Most simulations results are presented in supplementary table 1. The authors vary a few different parameters, and show the results in separate tables, what are the other parameters used in simulations?

I am not sure why the authors use spearman R instead of Pearson correlation R2, Pearson correlation is commonly used in assessing TWAS models. Pearson correlation R2 is more suitable here and this makes comparison with existing methods hard.

Where is the method section for "a new statistical method to compare TWAS effect sizes across cell types"? The authors indicated in the "online methods" but I didn't find it.

Reviewer #2 (Remarks to the Author):

Wang et al. present a novel method for training gene expression prediction model for TWAS based on only summary statistics. There were many assessments performed in simulated and real datasets. From current version, the EXPRESSO performed better in improving prediction accuracy and identifying larger number of novel/known loci than previous TWAS methods. However, I think more other assessments needed to carry out (as following mentioned). They also provide a CADRE pipeline for cell-type-aware drug repurposing, but it is hard to tell the significance of these findings since no gold-standard ground truth. Overall, the EXPRESSO method give a relatively clear assessment, and the assessment of this CADRE pipeline is largely insufficient. The paper give a clear writing. Here are my major concerns.

1. The algorithm of EXPRESSO is very similar with a previous method, PUMICE, they extended the PUMICE method to analyze summary statistics. In other words, EXPRESSO is only an extension of PUMICE. As the authors said, the extended method of EXPRESSO is very useful for parsing summary statistics, however, the algorithm per se has been used for TWAS in their earlier study, authors need to be clearly elucidated in the text for explaining the kinship between EXPRESSO and PUMICE, which enhance the readers understanding.

2. Why the authors did not compare the performance of EXPRESSO with other two widely-used summary statistics-based methods S-PrediXcan (for single-tissue) and S-MultiXcan (for multi-tissues)? I hope to see this benchmarking in simulation and real data.

3. Although the EXPRESSO is assessed and benchmarked by using bulk-based data, what performance of EXPRESSO in single-cell QTL data compared with other TWAS methods? Different sparsity and cell number of each cell type in single-cell data whether influence the performance of EXPRESSO? I hope more detailed assessments on cell-type-level effects should be given.

4. For the section on TWAS identifies autoimmune diseases-relevant cell types, the authors used 14 autoimmune diseases to enrich with seven immune cell types included in sc-eQTLGen, and they explained their results by using supportive evidence for reported studies. In this way, this enrichment analysis per se is awfully unfair. These diseases are immune-related traits defined by plenty of previous immunity-based studies. The authors conducted an analysis of the links between these diseases and immune cell types is highly inflated. Randomly selected genes from these diseases to do this analysis may yield significant immune cell types, which easily find some supporting biological

evidence from previous studies. For this part analyses, I can not get any convinced evidence. If added other diverse cell types like neuron sc-eQTLs, can it still perform well, or results remain unchanged?

5. For the section on cell type-level TWAS identifies and fine maps novel loci in disease-relevant cell type, the authors used the total number of all identified loci, novel loci, and known loci as metrics to assess the favorable performance of EXPRESSO with other TWAS methods. All number of all loci, novel loci, and known loci identified by EXPRESSO are higher than other tools. Is the proportion of identified novel loci by EXPRESSO higher than other TWASs ? Does this means the more identified loci the better? Are that more false positive loci identified by EXPRESSO?

6. Authors said "Many of those novel putatively causal genes have strong biological relevance in disease-relevant cell types (Supplemental Table S13)". This sentence is unclear. Are the enriched pathways associated with disease-relevant cell types? Why pathway relevance for these given cell types? Authors should give detailed indications.

7. For the section of cell-type aware computational drug repurposing using TWAS based on scRNA-seq data, authors gave a τ score based on the CMap database for indicating that the molecule will normalize the gene expression profile associated trait. They strict the drug repurposing analysis to six traits with known indications. However, only one general figure (Figure 5) to show the compared results between cell-type CADRE and those whole-blood RNA-seq dataset. They said that CADRE pipeline could yield a much lower average τ scores than that from whole-blood-based TWASs. These results are less convinced. I can not tell the advantage of CADRE for identifying trait-relevant cell type-specific drugs. For each disease, whether CADRE remain obtain lower τ scores? I hope the authors can do more detailed assessments on CADRE in a cell-type-specific manner, not just treated as a pseudo-bulk.

RESPONSE TO REVIEWERS

Reviewer #1 (Remarks to the Author):

1. Wang et al developed a new method to use summary statistics from eQTL mapping studies to build expression models. The unique feature is that it can leverage epigenomic information. The paper further developed a cell type aware drug repurposing pipeline. For the TWAS prediction model building method, I have some concerns about missing details, the evaluation of this method and the contribution of the method on real data analysis.

My major concern is that the study did not clearly differentiate from current methods that use summary statistics data to build prediction models. Using summary statistics to build prediction models for TWAS have been reported before (SUMMIT Nature communications 2022, OTTER Nature communication 2023), but they are not able to incorporate epigenetic information. Thus, I think the main advancement of the method is being able to use epigenetic/3D-genome information on top of using summary statistics to build the model, however, this has not been described clearly in the manuscript. Instead, most analysis results presented in the paper emphasized on the ability of using summary statistics, e.g., allows one to use much larger sample size, and cell type specific models are informative than tissue-level models. This is not related to the major advancement claimed by the paper (include epigenetic/3D genome in model) but solely being able to use summary statistics.

RESPONSE: Thank you for the suggestion! In the revised manuscript, we further clarified the major contributions of the proposed method (**page 11, paragraph 1**). First, as the reviewer suggested, EXPRESSO is the first method that can integrate eQTL summary statistics, 3D genome, and epigenetic information for predicting gene expressions. Second, we also propose a method named pseudo variable selection (PVS) for selecting tuning parameters. PVS is very useful as it is often difficult to identify validation data to tune hyperparameters when there is only eQTL summary statistic available. Besides, the application to single cell eQTL data for TWAS is novel. To our knowledge, this is the first application of TWAS using single cell eQTL data from immune cell types. We demonstrate how the mediating effects of gene expression vary across different cell types and identify a number of novel cell type specific target genes for autoimmune diseases.

We added the comparison with SUMMIT and OTTERS in the revised manuscript (**page 6**). OTTERS does not propose new methods for predicting gene expressions. Instead, it uses SDPR, PRS-CS, lassosum, and pruning and thresholding (P+T) methods to predict gene expressions with eQTL summary statistics as input and combine the TWAS results based on different prediction models via Cauchy combination method. In real data and simulation analysis, we compare the gene expression prediction accuracy of PRS-CS, SDPR, lassosum, and P+T. We observe that our method EXPRESSO-PVS outperforms all the other methods in terms of gene expression prediction accuracy, the proportion of significant model (**Supplementary table 7**), and TWAS power. When combining TWAS p-values from EXPRESSO-PVS with other methods using Cauchy's combination method, we can further improve TWAS power (**Supplementary table 10 and 11**). For sc-eQTL analysis, we compare the prediction accuracy for different cell types in **Supplementary Table 8** and the numbers of TWAS significant loci in **Supplementary Table 14**. We also added simulation comparisons (**Supplementary Methods; Supplementary Tables 1-5**).

As we explain in the response to comment 2 below, we attribute the improvement of EXPRESSO-PVS to its capability of incorporating 3D genome and epigenetic annotation as well as the PVS method to more effectively select tuning parameters.

2. More specifically, lacking information about the choice of epigenetic annotation. The authors used four

annotations tracks from ENCODE. Is this a universal choice? Why these four annotations are chosen and how do different annotations affect the performance of the method?

RESPONSE: Thank you for the comment! We choose the epigenomic annotation tracks (i.e., H3K27ac mark, H3K4me3 mark, DNase hypersensitive mark, and CTCF mark) because of their importance in gene transcription regulation and their broad availability across different tissues and cell types. To further evaluate how these annotations help improve prediction accuracy, we perform simulations without annotations (**Supplementary Methods**). In these new simulations and analyses, the probability of a variant being causal and the effect size distributions do not depend on the annotations. EXPRESSO-PVS (EXPRESSO using PVS for tuning parameter selection) still consistently outperforms other competing methods. We attribute the improvement of EXPRESSO-PVS in these scenarios to the PVS method. Another line of evidence supporting the utility of epigenetic and 3D genome data comes from applied data analysis of sc-eQTLGen. We observe that the advantage of EXPRESSO-PVS over alternative methods are bigger for cell types with matched 3D genome and epigenetic data. The advantage becomes smaller (yet remains) when we have to rely on the proxy cell types (i.e., the cell types with similar transcriptomic profiles) for annotation information.

As suggested by the reviewer, we also investigate how different annotations influence the result. We examine the distribution of various window sizes (w) and mitigation parameter (ϕ , which reduces the penalty parameter for essential variants) among the final EXPRESSO-PVS models. As shown in panel (A) of **Supplementary Figure 3**, 3D genome-informed regions (loop, TAD, domain, pcHi-C) were chosen 47.27% of the time. As a comparison, 1 million basepair window, as the default choice by many other methods, is only chosen 31.28% of the time. For the mitigation parameter, the most frequent choice is $\phi = 1/6$ (31.01%), which yields a much smaller penalty term to essential predictors. It suggests that assigning smaller penalty to essential variants that overlap functional genomic annotations improves prediction accuracy. These empirical comparisons showcase the utility of functional annotation for improving prediction accuracies.

3. The author mentioned Hi-C is also used, is this used as a separate class of essential variants? The authors indicated there is another tuning parameter w for Hi-C defined regions. How is it incorporated into the model? The given likelihood function does not have w . Similar to the questions or epigenetic data defined essential variants, how the choice of loop, TAD, and promoter regions affect results? Do the authors suggest to use all these categories to annotate or one category is more useful than the others?

RESPONSE: Thank you for the comment! We agree with the reviewer that the window size is not an explicit parameter in the model but have clarified that in the revised manuscript. We consider different choices of windows that may harbor regulatory variants as a tuning parameter, including windows defined by linear intervals (250kb and 1mb) and the ones defined by 3D genomes (i.e., loop, TAD, domain, and promoter capture Hi-C (pcHi-C) region). To fit the model, we first minimize the loss function for each choice of window. We then use nested cross validation and PVS to determine the window (as a hyper-parameter) that minimizes the loss function. We have revised our manuscript to make it clearer (**page 5, paragraphs 2-3**). As we responded to comment 2, a large fraction of models choose 3D genome-based windows and mitigation factors that minimize the penalty for essential predictors. These choices maximize prediction accuracy, which demonstrate the utility of annotations.

4. I think being able to account for functional annotation is a major contribution, but how much this information help to improve power, prediction accuracy etc is not clearly shown.

RESPONSE: Thank you for the comment! As our response to comment 2, we perform additional simulations and analyses without annotation information (**Supplementary Methods**). In this case, EXPRESSO-PVS still

outperforms all other methods, which demonstrate PVS, as a method for selecting tuning parameters without external validation data, helps improve the prediction accuracy. Moreover, in our analysis of sc-eQTLGen data, for cell types without matched 3D genome or epigenetic data, the improvement over alternative methods become smaller (but still remains). It also helps demonstrate the advantage of EXPRESSO-PVS when annotation information is incorporated. Finally, as we responded to comments 2 and 3, a large fraction of prediction models choose to include 3D genome or epigenetic annotations as models including these annotations achieve maximal prediction accuracy.

5. The authors uses a polygenic model in simulations and have two categories of variants with different effect size distributions. I think the current observation is that most genes have a few eQTLs and therefore sparse models should be included in simulations. And how robust it is to model misspecification, e.g., when there are more than two categories of variants effects size distribution?

RESPONSE: Thank you for the suggestion! In the revision, we add additional scenarios where variants in four different functional categories have their respective effect size distributions (**Supplementary Methods**). The effect sizes are determined by the heritability explained by each functional category and the number of variants in that functional category. We simulate eQTL effect sizes based on a mixture of four components, representing four different functional categories, including the H3K27ac mark, H3K4me3 mark, DNase hypersensitive mark, and CTCF mark. Variants in different categories have different effect size distributions. For variants in functional category j , we simulate their effect sizes according to

$$\beta_{epi(j)} \sim MVN \left(0, \frac{h_{epi(j)}^2}{M_{epi(j)}} \mathbf{I} \right)$$

$h_{epi(j)}^2$ is the heritability explained by the variants in annotation track j and $M_{epi(j)}$ is the number of variants that belong to annotation track j , and \mathbf{I} is an identity matrix. The proportions of gene expression heritability explained by essential variants in different categories follow values reported by an earlier study¹. For example, if the heritability explained by all essential variants is 0.05, we set the heritability explained by H3K27ac mark, H3K4me3 mark, DNase hypersensitive mark, and CTCF mark to be 0.0208, 0.0146, 0.0121 and 0.0025. Under this multiple component mixture model, the EXPRESSO model would be a mis-specified. Yet, as we show in **Supplementary Tables 1-5**, the advantage of EXPRESSO remains.

6. The authors says "It is clear from this comparison that EXPRESSO already outperforms methods based on individual level data, when they all analyze the same set of individuals." Does this increase in power comes from using epigenetic data? Compared with individual level methods EpiXcan and MOSTWAS (Bhattacharya et al 2021 Plos Genetics) which can also take multiomics data, where does this increase in power comes from? Like in the above comments, I think comparing with methods that can also account for functional annotation is a more direct comparison to help the readers understand the advancement in the methodology.

RESPONSE: Thank you for the suggestion! The increase in power comes from two parts: 1) using epigenetic and 3D genomic information, and 2) using the pseudo variable selection method for selecting tuning parameters. We already compared with EpiXcan in simulation and real data analysis of the original manuscript. EpiXcan integrates epigenetic but not 3D genome information. It generates priors from Roadmap chromHMM annotation and uses quadratic Bezier functions to rescale SNP priors to penalty factors which are used in a weighted elastic net regression. However, it is computationally expensive to obtain a rescaling equation and currently the authors only used 8 representative genes to pick an optimal Bezier function. It is less accurate in our comparisons possibly because it does not incorporate 3D genome information and the use of Bezier function may be too restrictive. The method also depends on the individual level data. In practice, it can have much lower power than EXPRESSO which can analyze summary statistics from much bigger datasets.

As we responded to comments 2 and 4, EXPRESO still performs better than other methods when the genetic effects and causality of eQTL variants do not depend on annotation. This improvement may be due to PVS which can more effectively select tuning parameters. m

In the revised manuscript, we further added the comparisons with DePMA and MeTWAS as proposed in the MOSTWAS paper (**page 5, paragraph 2**). DePMA and MeTWAS include eQTL SNPs or predicted expression (or other omics phenotypes) as mediators in the prediction. In our evaluations, they do not perform as well compared to many other methods (**Supplementary Tables 7, 10&11**). One possible reason is that the incorporated eQTL effects or predicted gene expression levels tend to be noisy and compromise the prediction accuracy. They also have to rely on individual level data. In practice, its prediction accuracy will be much lower compared to methods that can analyze much larger datasets of eQTL summary statistics.

7. Results for cell type specific TWAS is clearly showing that cell type specific GWAS is adding useful information to tissue level TWAS. However, I don't think this is directly related to how this proposed method brings new biological knowledge. Cell type specific TWAS has been performed (e.g <https://www.ncbi.nlm.nih.gov/pmc/articles/PMC9297655/>) so I think if the authors show that there are novel findings by comparing with results from existing analysis and these novel findings makes biological sense, then that would be a more direct evidence of the value of the method.

RESPONSE: Thank you for the comment! The suggested article does not perform TWAS. It performs colocalization analysis, which examines if the causal variants for SLE are the same as causal eQTL variants in the same locus. Colocalization focuses on loci already identified in GWAS and does not identify novel associated genes. Compared to colocalization analysis, TWAS allows us to identify novel gene level associations and also evaluate the mediating effect of gene expression levels (i.e., regulatory variants $\xrightarrow{\text{eQTL effects}}$ gene expression $\xrightarrow{\text{mediating effects}}$ SLE). As we demonstrate, EXPRESSO identified 36 (402) novel loci for SLE (and other autoimmune diseases), many of which are cell type specific and missed by TWAS using bulk tissues.

Other issues:

8. In general, I feel the the simulation results lack clear description. Most simulations results are presented in supplementary table 1. The authors vary a few different parameters, and show the results in separate tables, what are the other parameters used in simulations?

RESPONSE: Thank you for the suggestion! We add the comparisons in detail for all parameters setting in 216 scenarios. (**Supplementary tables 1-6**)

9. I am not sure why the authors use spearman R instead of Pearson correlation R2, Pearson correlation is commonly used in assessing TWAS models. Pearson correlation R2 is more suitable here and this makes comparison with existing methods hard.

RESPONSE: Thank you for the suggestion! We used Pearson correlation instead now (**Supplementary tables 7-8**) and we find the comparisons using Spearman correlation and Pearson correlation give nearly identical results as we use normalized gene expression levels.

10. Where is the method section for "a new statistical method to compare TWAS effect sizes across cell types"? The authors indicated in the "online methods" but I didn't find it.

RESPONSE: Thank you for comment! We have made it clear in the revised manuscript that it refers to the heterogeneity test (**page 15, paragraph 3**).

Reviewer #2 (Remarks to the Author):

Wang et al. present a novel method for training gene expression prediction model for TWAS based on only summary statistics. There were many assessments performed in simulated and real datasets. From current version, the EXPRESSO performed better in improving prediction accuracy and identifying larger number of novel/known loci than previous TWAS methods. However, I think more other assessments needed to carry out (as following mentioned). They also provide a CADRE pipeline for cell-type-aware drug repurposing, but it is hard to tell the significance of these findings since no gold-standard ground truth. Overall, the EXPRESSO method give a relatively clear assessment, and the assessment of this CADRE pipeline is largely insufficient. The paper give a clear writing. Here are my major concerns.

1. The algorithm of EXPRESSO is very similar with a previous method, PUMICE, they extended the PUMICE method to analyze summary statistics. In other words, EXPRESSO is only an extension of PUMICE. As the authors said, the extended method of EXPRESSO is very useful for parsing summary statistics, however, the algorithm per se has been used for TWAS in their earlier study, authors need to be clearly elucidated in the text for explaining the kinship between EXPRESSO and PUMICE, which enhance the readers understanding.

RESPONSE: Thank you for the comment! In the revised manuscript, we clarified the novelty of the proposed methods (**page 11, paragraph 1**). There are a few innovations in the proposed method. First, it extends PUMICE to summary statistics. To our knowledge, it is the first gene expression prediction method that integrates eQTL summary statistics with epigenetic and 3D genome data. Its performance is consistently higher than alternative methods and, in many cases, considerably higher. Second, we also propose a novel method pseudo variable selection (PVS) to select tuning parameters. PVS does not need external validation data to select tuning parameters, which is ideally suited for scenarios with only summary eQTL statistics available, as external validation data may not exist. In fact, with PVS, EXPRESSO can outperform PUMICE when analyzing data from the same of individuals but only need eQTL summary statistics as input (**Supplementary Tables 4, 5 & 7**).

2. Why the authors did not compare the performance of EXPRESSO with other two widely-used summary statistics-based methods S-PrediXcan (for single-tissue) and S-MultiXcan (for multi-tissues)? I hope to see this benchmarking in simulation and real data.

RESPONSE: Thank you for comment! EXPRESSO and all summary statistics-based methods compared use eQTL summary statistics to build gene expression prediction models. S-PrediXcan and S-MultiXcan takes gene expression prediction model as input and test for the association between phenotypes and predicted gene expression levels. They do not generate expression prediction methods.

3. Although the EXPRESSO is assessed and benchmarked by using bulk-based data, what performance of EXPRESSO in single-cell QTL data compared with other TWAS methods? Different sparsity and cell number of each cell type in single-cell data whether influence the performance of EXPRESSO? I hope more detailed assessments on cell-type-level effects should be given.

RESPONSE: Thank you for the suggestion! We compared EXPRESSO against other summary statistics-based method using sc-eQTLGen as training data and DICE as test data (**page 7**). The results for comparing prediction accuracy are in **Supplementary Table 8**. The results for comparing the number of identified TWAS loci are in **Supplementary Table 11**. Similar to the application to bulk eQTLs, EXPRESSO leads to a bigger number of

significant gene expression prediction models, a bigger prediction R^2 , and improved TWAS power, compared to alternative TWAS methods.

Based on the reviewer's suggestion, we also evaluate how the cell type proportion influence the average R^2 of the prediction accuracy and the number of models with significant R^2 . Interestingly, the prediction accuracy tends to be lower for rare cell types compared to more common cell types, possibly because sc-RNASeq can better capture the gene expression variability across individuals from more common cell types. The Pearson correlation between the number of significant models and cell type proportion is 0.27, and the Pearson correlation between the median prediction R^2 and cell type proportion is 0.33.

4. For the section on TWAS identifies autoimmune diseases-relevant cell types, the authors used 14 autoimmune diseases to enrich with seven immune cell types included in sc-eQTLGen, and they explained their results by using supportive evidence for reported studies. In this way, this enrichment analysis per se is awfully unfair. These diseases are immune-related traits defined by plenty of previous immunity-based studies. The authors conducted an analysis of the links between these diseases and immune cell types is highly inflated. Randomly selected genes from these diseases to do this analysis may yield significant immune cell types, which easily find some supporting biological evidence from previous studies. For this part analyses, I can not get any convinced evidence. If added other diverse cell types like neuron sc-eQTLs, can it still perform well, or results remain unchanged?

RESPONSE: Thank you for the comments! We are sorry that we do not fully understand the point the reviewer makes about the enrichment analysis. Yet, we hope that a clarification and additional analysis help answer this point. While it is well known that autoimmune disease is related to immune cell types, our knowledge of the actual cell types involved remains incomplete. It is also unclear whether the knowledge gained from animal studies are supported by human genetic evidence.

Cell type enrichment analysis is often applied to identify disease relevant cell types using genome-wide or transcriptome-wide association study results². Our analysis follows established method². We first create cell type specific gene sets as the genes with expression levels in the top 10th percentile in each cell type. We then use enrichment analysis to evaluate if cell type specific genes are enriched with TWAS hits from whole blood. The immune cell types enriched with TWAS hits are deemed relevant for disease.

We are confused about the reviewer's comments that the cell type enrichment analysis results are inflated. To demonstrate the validity of enrichment results and follow reviewer's suggestion (**page 8, paragraph 2; page 16, paragraph 1**), we use neuronal cell types as a negative control, as most of the disease are not related to brain cell types (except for multiple sclerosis). As expected, nearly all brain cell types are not enriched with TWAS signals from whole blood and immune cell types. This new analysis does not yield excessive false positive enrichments in disease irrelevant cell types (**Supplementary Table 13**) and help establish the validity of our enrichment analysis.

5. For the section on cell type-level TWAS identifies and fine maps novel loci in disease-relevant cell type, the authors used the total number of all identified loci, novel loci, and known loci as metrics to assess the favorable performance of EXPRESSO with other TWAS methods. All number of all loci, novel loci, and known loci identified by EXPRESSO are higher than other tools. Is the proportion of identified novel loci by EXPRESSO higher than other TWASs ? Does this means the more identified loci the better? Are that more false positive loci identified by EXPRESSO?

RESPONSE: Thank you for comment! All methods compared have well controlled type I errors as we establish in simulations. Also, all methods have well calibrated genomic control value when applied to real data (**Supplementary Table 12**). By assuming that a majority of reported hits in GWAS catalog is real, identifying a bigger number of reported loci in GWAS catalog (while having controlled type I error) demonstrates improved power. This approach has been used to compare different methods empirically. Besides, our method identifies more novel loci as well. We have made it clear in the revised manuscript (**page 8, paragraph 1**).

In addition, we also add another metric to support empirical power evaluation. Specifically, we use the mean value of chi-square statistic at known loci as a metric for empirical power comparison, as it estimates the non-centrality parameter of chi-square distribution³. In the revised manuscript, we added the comparisons of the median of square of Z-score at known loci in GWAS catalog in **Supplementary table 10**. TWAS based on EXPRESSO-PVS consistently yields bigger values of median Z-scores at known loci compared to other methods. When combining with other TWAS methods, Z-scores based on the combined p-values including EXPRESSO-PVS is further increased (**Supplementary Table 10**).

6. Authors said "Many of those novel putatively causal genes have strong biological relevance in disease-relevant cell types (Supplemental Table S13)". This sentence is unclear. Are the enriched pathways associated with disease-relevant cell types? Why pathway relevance for these given cell types? Authors should give detailed indications.

RESPONSE: Thank you for the comment! We apologize for the confusion! Cell type enrichment analysis is conducted independently of pathway enrichment analysis. For cell type enrichment analysis, we first identify cell type specific gene sets (**page 16, paragraph 1**) and then examine if the cell type specific gene sets are enriched with TWAS hits. In pathway enrichment analysis, we examine if a given pathway or GO term are enrichment with TWAS hits. The pathway may not depend on the cell type. We have revised the sentence to "Many of those novel putatively causal cell type specific associations have strong biological relevance" (**page 9, paragraph 2**).

7. For the section of cell-type aware computational drug repurposing using TWAS based on scRNA-seq data, authors gave a τ score based on the CMap database for indicating that the molecule will normalize the gene expression profile associated trait. They strict the drug repurposing analysis to six traits with known indications. However, only one general figure (Figure 5) to show the compared results between cell-type CADRE and those whole-blood RNA-seq dataset. They said that CADRE pipeline could yield a much lower average τ scores than that from whole-blood-based TWASs. These results are less convinced. I cannot tell the advantage of CADRE for identifying trait-relevant cell type-specific drugs. For each disease, whether CADRE remain obtain lower τ scores? I hope the authors can do more detailed assessments on CADRE in a cell-type-specific manner, not just treated as a pseudo-bulk.

RESPONSE: Thank you for comment and suggestions! In the revised manuscript, we provided the comparisons of τ scores across different diseases (**Supplementary Table 20**). CADRE yielded more negative average τ scores in five out of the six diseases compared, which shows that it can identify drugs that more consistently reverse gene expression levels across different cell types.

Besides, we also demonstrate the utility of cell type specific TWAS results in drug repurposing using an orthogonal approach based on drug target gene enrichment analysis. Specifically, if a drug is suitable for treating a given disease, the drug target genes may be enriched with disease risk genes. A promising drug may exhibit stronger enrichment with disease risk genes. To perform drug target enrichment analysis, we first generate gene sets for drug targeted pathways for each drug using DrugBank, a database that documents the mechanism of action for all approved drugs. We then perform drug target enrichment analysis using TESLA, a published

multiple regression-based method (**page 10, paragraph 4 and page 16, paragraph 4**). We compare the enrichment p-value between repurposing analysis using CADRE and bulk RNASeq data. Among the 50 drugs that were approved for treatment or in clinical trials, 12 are enriched with TWAS signals from disease relevant cell types and only 2 are enriched with hits from whole blood (two-sided Fisher's exact p-value 0.008). Besides, 88.9% of the 50 drugs are more enriched with cell type specific TWAS hits from disease relevant cell types than from whole blood. This replication analysis using an orthogonal method validates the importance of using TWAS hits from disease relevant cell types and confirms the improved performance of our cell type aware drug repurposing pipeline CADRE (**Supplementary Table 20**).

REFERENCES.

1. Finucane, H.K. *et al.* Partitioning heritability by functional annotation using genome-wide association summary statistics. *Nat Genet* **47**, 1228-35 (2015).
2. Boyle, E.A., Li, Y.I. & Pritchard, J.K. An Expanded View of Complex Traits: From Polygenic to Omnigenic. *Cell* **169**, 1177-1186 (2017).
3. Hayeck, T.J. *et al.* Mixed Model Association with Family-Biased Case-Control Ascertainment. *Am J Hum Genet* **100**, 31-39 (2017).

Reviewer #1 (Remarks to the Author):

I acknowledge the authors for providing additional analysis to address the major contributions of their study. I think these are useful analysis and are clearly presented by the authors. In my understanding, I think the main contribution of the study is that (1) it has a new parameter tuning algorithm to improve the prediction model built on summary statistics. (2) it can incorporate epigenetic information. I acknowledge these contributions, but I feel the improvement may be modest in practical settings.

On individual level data, as indicated by the authors, EXPRESSO's performance is similar to PUMICE, which is a previous method that can leverage epigenomic and 3D data. The difference between EXPRESSO and PUMICE lies in the procedure to choose the tuning parameter.

On summary statistics data, the author provides an analysis to show that without annotation the performance of EXPRESSO vs other methods (Supplementary Table 3). This is useful as it provides a baseline performance comparison between methods, when annotation is not available, or variants do not have annotations. The number of significant models is 76.2%, vs the second best 71.3%. on TWAS power, 60.9% vs 57.7%. This represents a 6.9% increase of the number of prediction models and 5.5% increase in TWAS power. The relative performance of different methods in difference settings are very different and the overall performance really depends on the which setting is the most frequent one. E.g. when looking at $n_SNPs > 2$, it seems that TWAS power becomes very similar, average power for EXPRESSO, 58.3%; average power of the second best, PRSCs is 58.9%.

I feel the power of TWAS is the most important metric here, as it can inform the readers in practice the benefits of applying the new method. It is a bit unfair to just compare EXPRESSO with the 4 methods in OTTERS separately, as I believe OTTERS combine the p values from those 4 methods, which should be better than using a single one. The results presented may provide a lower estimate of TWAS power for OTTERS.

When using annotations, the improvement of EXPRESSO compared with other methods is a little more obvious, but in different settings it is very different. I think the authors indicated the results are based on 1000 replicates, so it would be useful to know the standard error which is not available in current tables and figures. The power improvement in TWAS with annotation is 8.79% compared to the second best, on average. This is likely an upper bound of the performance improvement of EXPRESSO, as this estimation is based on simulations that assume the annotations correctly group the variants based on effect size, more than half of the causal variants have annotations and just one of the methods from OTTERS.

The authors replied to my previous comments that this paper, PMC9297655, didn't perform TWAS and they are the first to perform cell type specific TWAS. But please see Figure 5H. They performed cell type specific TWAS and also used a version of TWAS (CONTENT) that decompose the shared and cell type specific components. Their results identified 93 genes associated with SLE (73 novel). I also think the cell type specific part of CONTENT is more useful for the CADRE pipeline proposed in this study.

In summary, I think the authors clearly presented their analyses and the study makes several contributions, but I have still some reservations about the actual improvement in practice. A lot of emphasizes in the manuscript are on the ability of using summary statistics and the power gain comes from sample size increase, which seems distractive to me.

Reviewer #2 (Remarks to the Author):

Thanks to the authors that have done a plenty of work to answer my concerns. I feel good for this revised version. No more question.

REVIEWER COMMENTS

Reviewer #1 (Remarks to the Author):

1. I acknowledge the authors for providing additional analysis to address the major contributions of their study. I think these are useful analysis and are clearly presented by the authors. In my understanding, I think the main contribution of the study is that (1) it has a new parameter tuning algorithm to improve the prediction model built on summary statistics. (2) it can incorporate epigenetic information. I acknowledge these contributions, but I feel the improvement may be modest in practical settings.

On individual level data, as indicated by the authors, EXPRESSO's performance is similar to PUMICE, which is a previous method that can leverage epigenomic and 3D data. The difference between EXPRESSO and PUMICE lies in the procedure to choose the tuning parameter.

On summary statistics data, the author provides an analysis to show that without annotation the performance of EXPRESSO vs other methods (Supplementary Table 3). This is useful as it provides a baseline performance comparison between methods, when annotation is not available, or variants do not have annotations. The number of significant models is 76.2%, vs the second best 71.3%. on TWAS power, 60.9% vs 57.7%. This represents a 6.9% increase of the number of prediction models and 5.5% increase in TWAS power. The relative performance of different methods in difference settings are very different and the overall performance really depends on the which setting is the most frequent one. E.g. when looking at $n_SNPs > 2$, it seems that TWAS power becomes very similar, average power for EXPRESSO, 58.3%; average power of the second best, PRScs is 58.9%.

RESPONSE: Thank you for the comment! Previous studies have shown that gene expression levels tend to have sparse genetic architectures, meaning that the number of causal variants is usually small¹. The GTEx study² also showed that very few genes have more than 4 independent eQTL SNPs. We also showed in real data analysis that incorporating annotation information consistently improves gene expression prediction accuracy.

As a result, scenarios with a large number of causal variants and scenarios where annotation information does not work are uncommon and also the worst-case scenarios for EXPRESSO. We include these scenarios in the simulation for completeness. Importantly, in these worst-case scenarios, EXPRESSO-PVS still performs better than or at least comparably to alternative methods, which demonstrate the robustness of methods and the consistent improvement of power. For example, even in this worst-case scenario, the prediction accuracy of EXPRESSO-PVS is 6.36% higher over PRScs. PRScs, in contrasts, only improves the accuracy by 1.57% and 2.02% over the third (SDPR) or fourth best methods (LASSOSUM).

Under more realistic scenarios, e.g., when the number of causal variant equals to 2, without annotation information, the TWAS power of EXPRESSO-PVS is 68.7% which is substantially higher than the second-best method PRScs (55.7%). We also added the scenario when the number of causal variant equals to 4. The power of EXPRESSO-PVS is 65.0% which is still higher than PRScs (59.2%) without annotation information. With annotation information added to the model, the improvement of EXPRESSO gets even bigger (69.0% vs 53.2%).

Also, as shown in real data analysis, incorporating annotation information improves prediction accuracy. It also leads to the identification of a greater number of known loci and a larger mean chi-square test statistics at known loci (while maintaining calibrated genomic control values), both of which are commonly used benchmarks for empirical power evaluation. For example, when we used the eQTLGen dataset for training, EXPRESSO-PVS improves the prediction accuracy by 10% when compared to PRScs. PRScs only has 1.05% and 2.47% improvement over SDPR and LASSOSUM, the third and fourth best methods respectively. When training the model with sc-eQTLGen data from 7 cell types, EXPRESSO-PVS consistently increases the prediction accuracy by at least 15% over the second-best method PRScs. On the other hand, the improvement of PRScs over SDPR and LASSOSUM is only 6.20% and 1.93%.

2. I feel the power of TWAS is the most important metric here, as it can inform the readers in practice the benefits of applying the new method. It is a bit unfair to just compare EXPRESSO with the 4 methods in OTTERS separately, as I believe OTTERS combine the p values from those 4 methods, which should be better than using a single one. The results presented may provide a lower estimate of TWAS power for OTTERS.

RESPONSE: Thank you for the comment! OTTERS use Cauchy combination method to combine the p-values of different TWAS methods which yields further improvement of TWAS. As we responded to the previous comments, in the most likely scenarios (i.e., when the number of causal variants is small and when annotation helps), EXPRESSO-PVS substantially outperforms each method and in the least favorable scenarios, EXPRESSO-PVS still performs comparably. As a result, EXPRESSO-PVS outperforms each method that OTTERS combines. Importantly, adding EXPRESSO-PVS to the set of methods that are combined, we can identify many more loci (686 vs 488), more known loci (259 vs 221) and higher mean χ^2 statistics at known loci (36.14 vs 35.84), which demonstrate the value of EXPRESSO.

When using annotations, the improvement of EXPRESSO compared with other methods is a little more obvious, but in different settings it is very different. I think the authors indicated the results are based on 1000 replicates, so it would be useful to know the standard error which is not available in current tables and figures. The power improvement in TWAS with annotation is 8.79% compared to the second best, on average. This is likely an upper bound of the performance improvement of EXPRESSO, as this estimation is based on simulations that assume the annotations correctly group the variants based on effect size, more than half of the causal variants have annotations and just one of the methods from OTTERS.

RESPONSE: Thank you for the comment! The scenario that the reviewer raised is when gene expression prediction models are trained using with $n=20,000$, mimicking eQTLGen data, the largest eQTL dataset. At that sample size, the prediction accuracies for all methods start to saturate. Importantly, the TWAS power of EXPRESSO-PVS is still 8.79% higher than PRSCs. In the same scenario, EXPRESSO-PVS also improves the prediction accuracy and the proportion of significant models by 11.59% and 11.45% (Supplementary Table 1). Moreover, in our previous revisions, according to the reviewer's suggestion, we simulate scenarios when the annotation information is not accurate or when the variant effects are not dependent on annotations. As discussed above, in those worst-case scenarios, EXPRESSO-PVS still outperforms other methods. The improvement is also evident in real data analysis.

We also note that most of the bulk and single cell eQTL datasets have sample sizes in the hundreds. For those sample sizes, the advantage of EXPRESSO-PVS is much bigger. Compared to the second-best methods (PRSCs or SDPR), EXPRESSO-PVS leads to 16.80% increase in prediction accuracy, 15.77% increase in the number of significant models, and 18.46% increase in TWAS power (Supplementary Table 4).

The authors replied to my previous comments that this paper, PMC9297655, didn't perform TWAS and they are the first to perform cell type specific TWAS. But please see Figure 5H. They performed cell type specific TWAS and also used a version of TWAS (CONTENT) that decompose the shared and cell type specific components. Their results identified 93 genes associated with SLE (73 novel). I also think the cell type specific part of CONTENT is more useful for the CADRE pipeline proposed in this study.

RESPONSE: Thank you for pointing this out! We apologize for the confusion. We have now compared our findings with theirs. We note that the two studies use different datasets to train the expression prediction model and applied them to different GWAS datasets. So, we focus on the novel loci we identify using results from five cell types that can be matched between the two studies, including B cells, classical monocytes, natural killer cells, CD4 T cells, and CD8 T cells. Our results identified a total of 98 unique genes, including 90 novel genes that are not identified by Perez et al³ and 60 novel gene outside the 1mb window of GWAS catalog identified loci. Furthermore, we extended our comparisons to rheumatoid arthritis (RA) and Crohn's disease (CD). We identified 54 novel genes for RA and 97 novel genes for CD.

In summary, I think the authors clearly presented their analyses and the study makes several contributions, but I have still some reservations about the actual improvement in practice. A lot of emphasizes in the manuscript are on

the ability of using summary statistics and the power gain comes from sample size increase, which seems distracting to me.

RESPONSE: Thank you for the comment! As shown in real data by us and others, gene expression tends to have fewer independent eQTLs and incorporating annotation information consistently improves prediction accuracy in real data. As such, scenarios with many causal variants and scenarios where annotation information fails to help are uncommon and the worst-case scenarios for EXPRESSO-PVS. In those scenarios, EXPRESSO-PVS still consistently outperforms competing methods. Importantly, the magnitude of the improvement of EXPRESSO-PVS over the second-best method is often bigger than the improvement of the second best over alternative methods. Moreover, EXPRESSO outperforms each method in OTTERS individually. When combined with other methods in OTTERS using Cauchy combination method, it leads to substantial further improvement in power. We hope that this establishes the value of EXPRESSO when used alone or combined with other methods.

REFERENCE.

1. Mogil, L.S. *et al.* Genetic architecture of gene expression traits across diverse populations. *PLoS Genet* **14**, e1007586 (2018).
2. Consortium, G.T. The GTEx Consortium atlas of genetic regulatory effects across human tissues. *Science* **369**, 1318-1330 (2020).
3. Perez, R.K. *et al.* Single-cell RNA-seq reveals cell type-specific molecular and genetic associations to lupus. *Science* **376**, eabf1970 (2022).

Reviewer #1 (Remarks to the Author):

The authors have adequately addressed my questions. I don't have other comments.

Reviewer #1 (Remarks to the Author):

The authors have adequately addressed my questions. I don't have other comments.

We thank the reviewers for their constructive and helpful comments, which have significantly improved the quality and clarity of our manuscript.